# Reward-Guided Speculative Decoding for Efficient LLM Reasoning

**Baohao Liao** [* 1]  **Yuhui Xu** [* 2]  **Hanze Dong** [* 2]
**Junnan Li** [2]  **Christof Monz** [1]  **Silvio Savarese** [2]  **Doyen Sahoo** [2]  **Caiming Xiong** [2]

## Abstract

We introduce Reward-Guided Speculative Decoding (RSD), a novel framework aimed at improving the efficiency of inference in large language models (LLMs). RSD synergistically combines a lightweight draft model with a more powerful target model, incorporating a controlled bias to prioritize high-reward outputs, in contrast to existing speculative decoding methods that enforce strict unbiasedness. RSD employs a process reward model to evaluate intermediate decoding steps and dynamically decide whether to invoke the target model, optimizing the trade-off between computational cost and output quality. We theoretically demonstrate that a threshold-based mixture strategy achieves an optimal balance between resource utilization and performance. Extensive evaluations on challenging reasoning benchmarks, including Olympiad-level tasks, show that RSD delivers significant efficiency gains against decoding with the target model only (up to $4.4\times$ fewer FLOPs), while achieving significant better accuracy than parallel decoding method on average (up to +3.5). These results highlight RSD as a robust and cost-effective approach for deploying LLMs in resource-intensive scenarios. The code is available at https://github.com/BaohaoLiao/RSD.

## 1. Introduction

Scaling laws are widely recognized by the machine learning community as a foundational principle for the development of large language models (Hestness et al., 2017; Kaplan et al., 2020; Hoffmann et al., 2022). They emphasize that increasing both model size and dataset scale leads to improved loss reduction and, consequently, enhanced generalization

capabilities. When data and model size are scaled to extraordinary levels, performance can reach unprecedented heights. Large models demonstrate remarkable capabilities across diverse tasks, showcasing robust generalization and advanced reasoning skills (Brown et al., 2020; Hurst et al., 2024; Anthropic, 2024; Team et al., 2024).

These advancements result in high computational and economic costs. While training is resource-intensive, inference at scale is even costlier, requiring vast computational infrastructure and energy to serve billions of queries (Patterson et al., 2021). The exponential growth in inference costs makes it a key challenge for large model deployment, highlighting the need for efficient techniques (Frantar et al., 2022; Lin et al., 2024; Xu et al., 2023; Sun et al., 2023; Zhang et al., 2023b; Li et al., 2024a; Xu et al., 2024; Liao & Monz, 2024) to reduce energy use and ensure scalability.

Specifically, sequential token generation in large LLMs incurs significantly higher computational costs compared to smaller models. This increased latency can hinder their deployment in real-time or high-throughput applications. To address this issue, parallel decoding techniques, such as speculative decoding, have emerged as effective solutions (Leviathan et al., 2023). Speculative decoding operates by leveraging a smaller, lightweight model to generate candidate outputs, which are then validated and refined by the larger model. This approach significantly reduces the number of decoding tokens required by the larger model, thereby accelerating the overall process. The smaller model serves as a guide, proposing sequences that the larger model can confirm or adjust, leading to faster inference without compromising quality. Furthermore, speculative decoding ensures efficiency by maintaining high-quality outputs through careful calibration of the smaller model. By aligning the smaller model's predictions with the larger model's capabilities, this method minimizes discrepancies and enhances reliability during inference.

Despite advancements in parallel decoding, speculative decoding remains underutilized for complex reasoning tasks, particularly multi-step generation. A key limitation is the strict *unbiasedness* requirement, which ensures the final token distribution matches the large model's but restricts flexibility in exploring diverse completions (Holtzman et al.,

---

[*]Equal contribution [1]Language Technology Lab, University of Amsterdam [2]Salesforce AI Research. Correspondence to: Yuhui Xu <yuhui.xu@salesforce.com>.

*Proceedings of the $42^{nd}$ International Conference on Machine Learning*, Vancouver, Canada. PMLR 267, 2025. Copyright 2025 by the author(s).

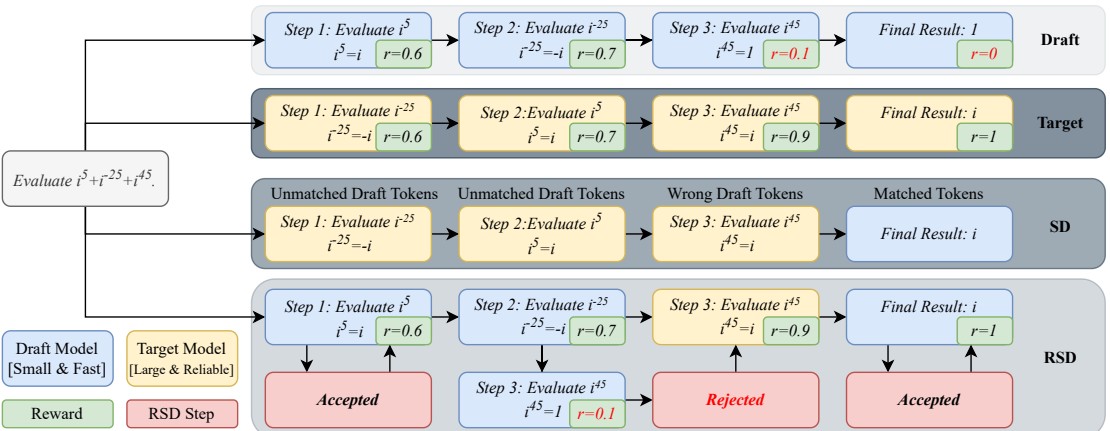

*Figure 1.* **Reward-Guided Speculative Decoding (RSD).** This diagram illustrates how RSD improves upon standard speculative decoding (SD) by incorporating reward-guided selection. SD strictly enforces exact token matching between the draft and target model, leading to unnecessary computations when mismatched tokens are discarded. In contrast, RSD evaluates draft outputs based on reward signals $r$ and selectively refines them, reducing reliance on exact matching and improving efficiency. The process starts with a small and fast draft model generating preliminary results, followed by a larger and more reliable target model verifying and refining predictions. Darker background regions indicate higher computational costs, showing how SD wastes resources on rejected tokens, whereas RSD reduces unnecessary steps by accepting useful draft outputs even when they do not exactly match, balancing efficiency and accuracy.

2020). While unbiasedness maintains theoretical fidelity, it often reduces efficiency, especially when the draft model diverges from the large model. High-quality tokens (e.g., those favored by a process reward) may still be rejected if their probabilities under the large model are too low, leading to wasted computation and negating potential speedups. This dependence inflates overhead and limits speculative decoding's benefits, particularly in long-trajectory reasoning tasks like math and coding. Allowing controlled *bias*, where the final distribution deviates slightly from the large model, can improve performance. If a draft token is correct but does not match the large model's distribution exactly, strict rejection is counterproductive. Reward-guided acceptance retains valuable partial solutions, reduces unnecessary queries, and can even surpass the large model's performance. Thus, more adaptive approaches are needed to balance efficiency and robustness, ensuring broader real-world applicability.

In this work, we introduce Reward-Guided Speculative Decoding (RSD), a novel framework that balances efficiency and accuracy by integrating computationally lightweight "draft" evaluations with reward-driven refinements from a more capable "target" model. Unlike traditional speculative decoding, which strictly enforces unbiasedness, RSD leverages reward signals to adaptively select high-value draft outputs rather than discarding mismatched tokens outright. The process begins with the draft model generating candidate steps, which are then evaluated using a reward function. Steps with sufficiently high reward scores are accepted to continue the reasoning trajectory, while lower-scoring steps trigger speculative corrections using the target model. As illustrated in Fig. 1, this adaptive mechanism

is robust against the distribution shifts issue between the draft and target models while optimizing resource allocation. By dynamically adjusting when to invoke the larger model, RSD significantly reduces unnecessary computations while maintaining or even surpassing the quality of traditional inference approaches. This approach is particularly well-suited for long-horizon reasoning tasks, where balancing computational cost and accuracy is critical.

**Contributions.** We propose Reward-Guided Speculative Decoding, a novel approach to accelerate LLM inference, particularly for reasoning tasks. It introduces an adaptive decoding framework that dynamically mixes outputs from a draft and target model, guided by a reward function that evaluates output quality at each step. This enables efficient, high-quality reasoning by constructing a flexible mixture distribution, $\mathbf{P}_{\text{RSD}}$, balancing efficiency and accuracy through reward-based weighting. RSD employs rejection sampling to selectively refine draft outputs, ensuring scalability. Theoretically, we derive optimal weighting strategies under computational constraints, maximizing efficiency without performance drop. Extensive experiments on GSM8K, MATH500, Olympiad Bench, GPQA, MMLU STEM, and GaoKao-2023-En show that RSD not only improves the reasoning accuracy up to 3.5 than SD on average, but also significantly reduces the inference computation with up to $4.4\times$ fewer FLOPs, compared to using the target model alone.

## 2. Reward-Guided Speculative Decoding

**Notations.** Let all tokens be embedded in Euclidean space. The prompt is represented as $x \in \mathbb{R}^{l \times d}$, and the response as

$y \in \mathbb{R}^{L \times d}$. The response $y$ can be further decomposed into a sequence of steps $[y_1, \cdots, y_n]$, which we denote as $y_{1:n}$.

For language models, we consider the iterative process of generating a sequence of steps $y_{1:n}$ given an input $x$ and a model $m$ (small/draft model) or $M$ (large/target model). At each step $i$, the **context** $z_i$ is constructed by combining the initial input $x$ with the sequence of previously generated outputs $y_{1:i-1}$, such that $z_i = [x, y_{1:i-1}]$.

Using this context, the next output $y_i$ is sampled from a conditional distribution: $y_i \sim \mathbf{P}_m(y_i|z_i)$ or $y_i \sim \mathbf{P}_M(y_i|z_i)$, where $\mathbf{P}_m$ corresponds to a small draft model $m$; $\mathbf{P}_M$ corresponds to a large target model $M$.

Our goal is to optimize the expected reward at each step with computation constraints. For each step $i$, we define a **reward function** $r(y_i|z_i) = r(y_i|x, y_{1:i-1})$, which evaluates the quality of the generated step $y_i$ within the sequence $y_{1:i}$ given prompt $x$. A higher reward $r(y_i|z_i)$ indicates a greater likelihood that the model output aligns with the desired response given the $x$ and $y_{1:i-1}$.

We consider the case that the expected reward achieved by the large model $M$ at each step satisfies:

$$\mathbf{E}_{y_i \sim \mathbf{P}_M} \left[ r(y_i|z_i) \right] \geq \mathbf{E}_{y_i \sim \mathbf{P}_m} \left[ r(y_i|z_i) \right], \quad (1)$$

i.e., the target model $M$ should outperform or at least match the draft model $m$ in terms of the expected reward at every step. Our analysis is based on the fact that the large model's predictions yield higher quality outputs, leveraging its capacity for complex reasoning and contextual understanding.

We define the distribution $\mathbf{P}_{\text{RSD}}$ as a dynamic mixture of $\mathbf{P}_m$ and $\mathbf{P}_M$, where the combination depends on the quality of the conditional output $y_i|z_i$. Specifically, we have:

$$\mathbf{P}_{\text{RSD}}(y_i|z_i) = w(y_i|z_i)\mathbf{P}_m(y_i|z_i) + v(y_i|z_i)\mathbf{P}_M(y_i|z_i),$$

where $w(\cdot)$ and $v(\cdot)$ are weighting functions that dynamically adjust based on the quality of the output $y_i|z_i$. Unlike $\mathbf{P}_m$, we assume $\mathbf{P}_M$ is sufficiently robust and reliable (also costs more); therefore, we set $v(y_i|z_i) = \nu$, where $\nu$ is a constant. This ensures that $\mathbf{P}_M$ always contributes to the mixture and is not rejected outright, as it acts as a stable fallback for handling low-quality outputs. In our approach, $w(y_i|z_i)$ is determined by a reward function $r$, such that:

$$w(y_i|z_i) = \omega_r(y_i|z_i) = \omega(r(y_i|z_i)),$$

where $r(y_i|z_i)$ measures the quality or preference for the conditional output $y_i|z_i$. The function $\omega(\cdot)$ maps $r(y_i|z_i)$ to a value in $[0, 1]$, reflecting the confidence in $\mathbf{P}_m$. For example, $r(y_i|z_i)$ could depend on factors like the accuracy or relevance of $y_i|z_i$, and $\omega(r(y_i|z_i))$ controls how much weight is assigned to $\mathbf{P}_m$ relative to $\mathbf{P}_M$. Because $w(y_i|z_i)\mathbf{P}_m(y_i|z_i)$ is an unnormalized distribution, the constant $\nu$ ensures proper normalization of the mixture.

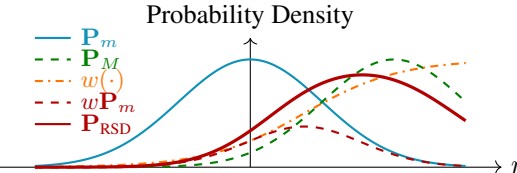

*Figure 2.* Illustration of Reward-Guided Speculative Decoding

When $r(y_i|z_i)$ is large, indicating a highly preferred or high-quality output, $\omega(r(y_i|z_i))$ approaches 1. In this case, $w(y_i|z_i)\mathbf{P}_m$ dominates the mixture, and $\mathbf{P}_{\text{RSD}}(y_i|z_i)$ primarily reflects the predictions of $\mathbf{P}_m$. This shows high confidence in the smaller model's ability to produce reliable results. Conversely, when $r(y_i|z_i)$ is small, $\omega(r(y_i|z_i))$ approaches 0. The mixture weight then shifts toward $\nu\mathbf{P}_M$, which allows the larger model $\mathbf{P}_M$ to dominate. Since $\mathbf{P}_M$ is assumed to be sufficiently robust and reliable, it compensates for the smaller model by effectively handling low-quality outputs, ensuring overall performance stability.

Fig. 2 depicts the mixture distribution $\mathbf{P}_{\text{RSD}}$, which is influenced by the quality of the output $y_i|z_i$. The blue curve represents the distribution $\mathbf{P}_m$, and the green dashed curve represents $\mathbf{P}_M$. The orange dotted line shows the weighting function $w(y_i|z_i)$, which adjusts the mixture between these two distributions. In regions where $y_i|z_i$ corresponds to high-quality outputs, the weighting function $w$ is elevated, placing more weight on $\mathbf{P}_m$ to prioritize efficiency. For low-quality outputs, $w$ decreases, shifting the emphasis toward $\mathbf{P}_M$ and thereby penalizing low-quality samples. The red curve represents the resulting mixture distribution $\mathbf{P}_{\text{RSD}}$, which adapts based on the quality of the output, demonstrating a more efficient sampling process for higher-quality outputs and a more penalized approach for low-quality ones. Regarding the efficiency, a large proportion of samples are drawn from $\mathbf{P}_m$, with a small part sourced from $\mathbf{P}_M$. Additionally, we maximize $\mathbf{E}_{y_i \sim \mathbf{P}_{\text{RSD}}} r(y_i|x, y_{1:i-1})$ to ensure that the model performs effectively in reasoning tasks.

## 2.1. RSD Algorithm

The Reward-Guided Speculative Decoding (RSD) algorithm operates by dynamically mixing the draft model $m$ and target model $M$ at each generation step, with the objective of balancing efficiency and quality. The algorithm leverages the reward function $r$ to guide this dynamic mixing, ensuring that necessary higher-quality steps are more likely to be generated by the target model $M$, while the draft model $m$ is used for cost-effective generation when possible. Below, we describe the key components of the algorithm . At each decoding step $i$, the algorithm follows these steps:

1. **Generate Draft Step:** The draft model generates a candidate $\hat{y}_i$ given the prompt and previous outputs.

2. **Compute Reward:** The reward function $r(y_i \mid z_i)$,

**Algorithm 1** RSD: Reward-Guided Speculative Decoding

**Input:** Prompt $x$, draft model $m$, target model $M$, process reward model $r$, acceptance criterion $\mathcal{A}_\omega$, EOS token $s$, max length $N$

Assign $y_{1:0} \leftarrow$ ""

**for** $i = 1$ **to** $N - 1$ **do**

    Generate draft step $\hat{y}_i \leftarrow m(x, y_{1:i-1})$

    Compute reward $r_i \leftarrow r(y_i | x, y_{1:i-1})$

    **if** $\mathcal{A}_\omega(r_i)$ **then**

        Accept the draft step $y_i \leftarrow \hat{y}_i$

    **else**

        Generate a target step $y_i \leftarrow M(x, y_{1:i-1})$

    **end if**

    **if** $s \in y_i$ **then**

        **break**

    **end if**

**end for**

**Output:** Response $y_{1:i}$

---

**Algorithm 2** Acceptance Criterion $\mathcal{A}_\omega$

**Input:** value $r \in \mathcal{R}$, weighting function $\omega : \mathcal{R} \to [0, 1]$

Compute weighting function $\omega(r)$

**if** $\omega(r) = 0$ or $1$ **then**

    $\mathcal{A}_\omega(r) = \omega(r)$

**else**

    Sample $u \sim \mathcal{U}(0, 1)$; $\mathcal{A}_\omega(r) = \mathbf{1}(\omega(r) \geq u)$

**end if**

**Output:** $\mathcal{A}_\omega(r)$

---

*Table 1.* Variants of Weighting Function $\omega(r)$ and their definitions

| Weighting Function | Definition |
|---|---|
| $\omega(r) = p$ | Constant $p \in (0, 1)$ |
| $\omega(r) = \mathbf{1}(r \geq \delta)$ | 1 if $r \geq \delta$, else 0 |
| $\omega(r) = \min(1, \max(0, r))$ | Clipping $r$ within [0,1] |
| $\omega(r) = \max(0, \frac{r}{1+r})$ | Sigmoidal transformation |
| $\omega(r) = \frac{1}{1+e^{-\alpha(r-\delta)}}$ | Logistic function |

where $z_i = [x, y_{1:i-1}]$, evaluates the step's quality.

3. **Apply Acceptance Criterion:** The reward $r_i$ is assessed using $\mathcal{A}_\omega(r_i)$. If accepted, $\hat{y}_i$ is used; otherwise, the target model $M$ generates a new step.

4. **Sample from Mixture Distribution:** Accepted steps come from $\mathbf{P}_m$, rejected ones from $\mathbf{P}_M$, dynamically balancing efficiency and accuracy.

5. **Repeat Until Termination:** Steps are generated until the EOS token appears or sequence reaches length $N$.

**Computational Efficiency.** The primary advantage of the RSD algorithm lies in its ability to combine the strengths of both the draft and target models while minimizing the computational cost. By using the draft model $m$ for most of the generation process, the algorithm reduces the computational burden compared to a strategy that always uses the target model $M$. The dynamic weighting function $w(r)$ ensures that the draft model is used when the generated steps are of sufficient quality, thereby maximizing efficiency.

Moreover, by employing rejection sampling, the algorithm only resorts to the more expensive target model $M$ when necessary, ensuring that high-reward steps are generated while keeping the overall cost low. This balance between cost and quality is particularly important in large-scale applications, where both efficiency and performance are critical.

**Formal Description of the Algorithm.** The RSD algorithm is formally described in Algorithm 1, and the acceptance criterion is outlined in Algorithm 2. In the following, we present the theoretical basis for the mixture distribution used in the algorithm. We also provide the final distribution of the proposed algorithm in Proposition 2.1. That means the expected reward from $\mathbf{P}_{\text{RSD}}$ lies within the reward bounds defined by $\omega_r \mathbf{P}_M(y|z)$ and $\mathbf{P}_M(y|z)$.

**Proposition 2.1.** *Given the Algorithm 1 and 2, for each step*

$y|z$. *Assume that $\omega_r(y|z) = \omega(r(y|z))$, the*

$$\mathbf{P}_{\text{RSD}}(y|z) = \omega_r(y|z)\mathbf{P}_m(y|z) + \nu\mathbf{P}_M(y|z), \quad (2)$$

*where $\omega_r(y|z)$ is the weighting function that adjusts the relative contributions of the draft model $\mathbf{P}_m$, and $\nu$ is the normalizing constant $\nu = 1 - \mathbf{E}_{\mathbf{P}_m}\omega_r$.*

### 2.2. Acceptance Criterion $\mathcal{A}_\omega$ and Weighting Function

**Proposition 2.2.** *Given the following assumptions:*

- *$\omega(r)$ is non-decreasing in $r$;*

- *$\mathbf{E}_{\mathbf{P}_M}[r(y|z)] \geq \mathbf{E}_{\mathbf{P}_m}[r(y|z)]$;*

*it follows that the expected value of $r(y|z)$ under the RSD induced distribution satisfies: $\mathbf{E}_{\mathbf{P}_{\text{RSD}}}[r(y|z)] \geq \mathbf{E}_{\mathbf{P}_m}[r(y|z)]$.*

The weighting function $\omega(\cdot)$ plays a crucial role in adjusting the mixture distribution. Several variants of the weighting function are considered. Table 1 provides a summary of different choices. Each of these variants provides new trade-off between draft model and target model. Intuitively, a binary function can maximize expected reward under a strict sampling budget constraint, a smooth weighting might in practice handle noisy reward model outputs more gracefully.

### 2.3. Optimal Weighting

In the following Proposition, we demonstrate that the optimal weighting function for maximizing reward under a constrained sampling budget is a binary step function, which assigns a weight of 1 only to high-reward outputs.

**Proposition 2.3.** *Given a constrained sampling budget $\nu = 1 - \mathbf{E}_{y \sim p_m}\omega_r(y|z) \leq \gamma$, $\gamma \in (0, 1)$, the optimal sampling*

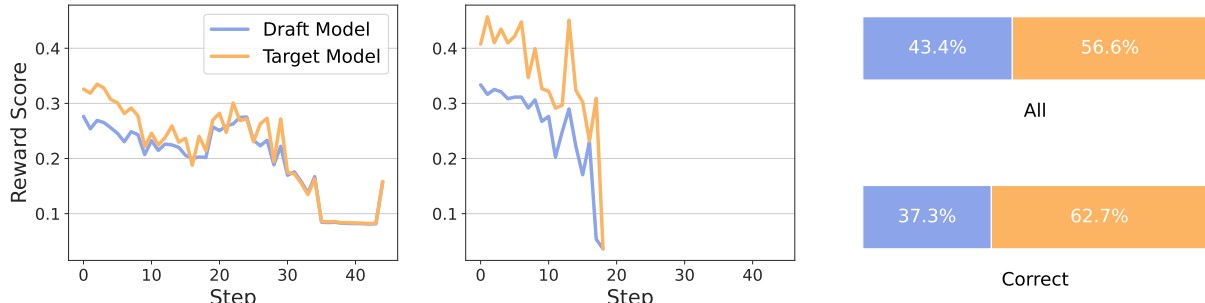

*Figure 3.* **Left**: A comparison of the reward scores for all questions generated by the draft model and the target model within the RSD framework. **Middle**: A focused comparison of the reward scores for correctly answered questions generated by the draft model and the target model in the RSD framework. **Right**: The winning rate (in terms of reward) comparison between the draft model and the target model, highlighting the proportion of cases where each model outperforms the other. RSD is configured with Qwen2.5-Math-1.5B-Instruct as the draft model, Qwen2.5-Math-7B-Instruct as the target model, and Skywork-o1-Open-PRM-7B as the PRM.

*strategy that maximize the reward is*

$$\omega_r(y|z) = \begin{cases} 1 & \text{if } r(y|z) \geq \delta_\gamma(z), \\ 0 & \text{if } r(y|z) < \delta_\gamma(z), \end{cases} \quad (3)$$

*where $\delta_\gamma(z)$ is the largest possible threshold that makes the function satisfy the constraint.*

Note that when a binary function is used, Algorithm 2 almost surely degenerates into a deterministic procedure, producing a definite acceptance or rejection outcome. In real implementation, $\delta_\gamma$ can be a hyper-parameter $\delta$.

### 2.4. Discussion

**Process Reward for Each Model.** As stated in Eq. (1), we expect the target model to have a higher reward. Fig. 3 confirms this on MATH500, showing that for correctly answered questions within RSD, $\mathbf{P}_M$ consistently outperforms $\mathbf{P}_m$ in reward (**Middle** figure).

**Comparison with the reward of $\mathbf{P}_M$.** Since $\mathbf{P}_{\text{RSD}}$ is a mixture of $\omega_r \mathbf{P}_m$ and $\mathbf{P}_M$, its reward is a weighted sum of their expected rewards. Thus, $\mathbf{P}_{\text{RSD}}$ can exceed $\mathbf{P}_M$ in expected reward when we use an aggressive $w$ that only assigns 1 to high-reward regions. In cases where $\omega_r \mathbf{P}_m$ has a higher expected reward than $\mathbf{P}_M$, $\mathbf{P}_{\text{RSD}}$ also achieves a higher reward. We also notice this phenomenon empirically.

**General Weighting Function and SD.** Notably, *reward-based* weighting functions $\omega_r$ are not the only option. An alternative approach explicitly defines the weighting function in terms of the likelihood ratio:

$$w(y \mid z) = \min\left(1, \alpha \frac{\mathbf{P}_M(y \mid z)}{\mathbf{P}_m(y \mid z)}\right),$$

where $\alpha > 0$ is a hyperparameter controlling how quickly the method transitions from the draft model to the target model. This formulation is algorithmically similar to speculative decoding, with $\alpha$ as a tunable parameter and $\mathbf{P}_M$

serving as the alternative distribution. In standard speculative decoding, one often uses $\mathbf{P}_M - \mathbf{P}_m$ to propose alternative generations, which allows unbiased final distribution with $\alpha = 1$. Intuitively, if a token (or reasoning step) is *significantly less likely* under the large model than under the draft model (i.e., if $\frac{\mathbf{P}_M}{\mathbf{P}_m}$ is small), it may indicate a suspicious scenario that the draft model is unable to handle effectively. More generally, *hybrid* approaches can combine both the reward function $r$ and the ratio $\frac{\mathbf{P}_M}{\mathbf{P}_m}$, for example: $w(y|z) = \min\left(1, \beta\, r(y|z)\, \frac{\mathbf{P}_M(y|z)}{\mathbf{P}_m(y|z)}\right)$, or by normalizing these two quantities in a differentiable manner. These ratio-based or hybrid weighting functions may help address situations where (i) the reward model is noisily correlated with true correctness or (ii) the small and large model distributions diverge substantially.

Hence, the weighting function in Reward-Guided Speculative Decoding can be flexibly designed to incorporate *process rewards*, *likelihood ratios*, or any combination thereof, as long as it meets the non-decreasing requirement in $r$ (when appropriate) and remains bounded within $[0, 1]$. This flexibility allows RSD induced algorithm to be adapted to different practical constraints (e.g., distribution mismatch, reward model availability/accuracy) while still reaping the efficiency gains of speculative decoding.

## 3. Empirical Results

**Models.** To assess the performance of RSD, we employ both general-purpose and math-focused LLMs as our target and draft models, specifically Qwen-2.5 (Yang et al., 2024a), Llama-3 (Dubey et al., 2024), and Qwen-2.5-Math (Yang et al., 2024b). Our system utilizes Skywork-o1-Open-PRM (o1 Team, 2024) as the process reward model (PRM), as it was the most advanced open-source PRM available during our experiments (Zheng et al., 2024). The reward score ranges from 0 to 1, the higher the better.

*Table 2.* Accuracy on reasoning benchmarks. In general, $\delta = 0.7$ offers a good trade-off between accuracy and efficiency for all tasks. $\delta^*$ is the optimized threshold for each task, as different tasks have a different complexity of reasoning. Refer to §B.1 for detailed results.

| Method | Target Model | Draft Model | PRM | Setting | MATH500 | GSM8K | GaoKao 2023 En | Olympiad Bench | GPQA Diamond | MMLU STEM | Avg. |
|---|---|---|---|---|---|---|---|---|---|---|---|
| *Math Model, Target and Draft: Qwen2.5-Math-Instruct, PRM: Skywork- o1-Open-PRM* | | | | | | | | | | | |
| Single Model | 7B | - | - | - | 83.2 | 95.7 | 66.8 | 41.2 | 32.8 | 71.8 | 65.3 |
| Majority Voting | - | 1.5B | - | maj@16 | 79.0 | 88.9 | **69.9** | **45.5** | 27.3 | 65.9 | 62.3 (-3.0) |
| Best-of-$N$ | - | 1.5B | 7B | $N=16$ | 82.2 | 93.3 | 69.4 | 44.9 | 27.3 | 71.4 | 64.8 (-0.5) |
| SD | 7B | 1.5B | - | - | 83.4 | 95.6 | 67.3 | 40.6 | 28.8 | 72.0 | 64.6 (-0.7) |
| RSD | 7B | 1.5B | 1.5B | $\delta = 0.7$ | 82.6 | 94.5 | 68.8 | 39.6 | **38.4** | 71.4 | 65.9 (+0.6) |
| RSD | 7B | 1.5B | 7B | $\delta = 0.7$ | **84.6** | 95.5 | 68.3 | 42.1 | 33.8 | 72.3 | 66.1 (+0.8) |
| RSD | 7B | 1.5B | 1.5B | $\delta^*$ | 82.6 | 95.5 | 68.8 | 40.6 | **38.4** | 71.7 | 66.3 (+1.0) |
| RSD | 7B | 1.5B | 7B | $\delta^*$ | **84.6** | **95.8** | 68.3 | 43.6 | 34.3 | **72.6** | **66.5 (+1.2)** |
| Single Model | 72B | - | - | - | 85.6 | 95.8 | 73.0 | 48.4 | 42.4 | 85.8 | 71.8 |
| Majority Voting | - | 1.5B | - | maj@64 | 80.2 | 89.8 | 71.2 | 45.9 | 30.8 | 66.1 | 64.0 (-7.8) |
| Best-of-$N$ | - | 1.5B | 7B | $N=64$ | 82.4 | 94.5 | 68.6 | 44.3 | 27.8 | 72.4 | 65.0 (-6.8) |
| Majority Voting | - | 7B | - | maj@64 | **88.0** | 96.5 | 73.8 | 47.6 | 35.9 | 77.2 | 69.8 (-2.0) |
| Best-of-$N$ | - | 7B | 7B | $N=64$ | 86.2 | **97.2** | 71.4 | 44.4 | 36.4 | 75.4 | 68.5 (-3.3) |
| SD | 72B | 7B | - | - | 84.8 | 95.8 | 71.7 | 47.6 | 41.4 | 85.3 | 71.1 (-0.7) |
| RSD | 72B | 7B | 1.5B | $\delta = 0.7$ | 86.4 | 96.4 | 73.0 | 49.8 | 42.9 | 84.7 | 72.2 (+0.4) |
| RSD | 72B | 7B | 7B | $\delta = 0.7$ | **88.0** | 96.7 | **74.0** | **49.9** | 42.9 | 84.4 | 72.7 (+0.9) |
| RSD | 72B | 7B | 1.5B | $\delta^*$ | 86.6 | 97.0 | 73.2 | 49.8 | **43.9** | 85.2 | 72.6 (+0.8) |
| RSD | 72B | 7B | 7B | $\delta^*$ | **88.0** | 96.9 | **74.0** | **49.9** | 42.9 | **85.6** | **72.9 (+1.1)** |
| *General Model, Target and Draft: Qwen2.5-Instruct, PRM: Skywork- o1-Open-PRM* | | | | | | | | | | | |
| Single Model | 7B | - | - | - | 77.4 | 92.0 | 64.9 | 38.8 | 28.8 | 57.4 | 59.9 |
| Majority Voting | - | 1.5B | - | maj@16 | 66.4 | 82.1 | 56.9 | 28.7 | 27.3 | 67.2 | 54.8 (-5.1) |
| Best-of-$N$ | - | 1.5B | 7B | $N=16$ | 73.4 | 89.7 | 60.5 | 32.7 | 23.7 | 69.4 | 58.2 (-1.7) |
| SD | 7B | 1.5B | - | - | **77.8** | 91.8 | 63.1 | 39.1 | 26.3 | 56.2 | 59.1 (-0.8) |
| RSD | 7B | 1.5B | 1.5B | $\delta = 0.7$ | 73.6 | 90.8 | 64.2 | 39.0 | **31.3** | 71.6 | 61.8 (+1.9) |
| RSD | 7B | 1.5B | 7B | $\delta = 0.7$ | 75.0 | **93.3** | **66.2** | 39.9 | 20.2 | 61.8 | 59.4 (-0.5) |
| RSD | 7B | 1.5B | 1.5B | $\delta^*$ | 74.8 | 92.3 | 65.2 | 40.0 | **31.3** | **71.7** | **62.6 (+2.7)** |
| RSD | 7B | 1.5B | 7B | $\delta^*$ | 75.8 | **93.3** | **66.2** | **40.9** | 29.8 | 66.3 | 62.1 (+2.2) |
| *General Model, Target and Draft: Llama-3.1-Instruct, PRM: Skywork- o1-Open-PRM* | | | | | | | | | | | |
| Single Model | 8B | - | - | - | 49.4 | 83.9 | 41.3 | 14.5 | 20.2 | 39.1 | 41.4 |
| Majority Voting | - | 1B | - | maj@16 | 38.0 | 60.2 | 32.2 | 9.5 | 19.7 | 24.9 | 30.8 (-10.6) |
| Best-of-$N$ | - | 1B | 7B | $N=16$ | **52.6** | 74.8 | **45.7** | 14.4 | 14.1 | 31.0 | 38.8 (-2.6) |
| SD | 8B | 1B | - | - | 47.0 | 83.4 | 42.1 | 16.6 | 19.2 | 38.5 | 41.1 (-0.3) |
| RSD | 8B | 1B | 1.5B | $\delta = 0.7$ | 50.0 | 83.9 | 41.8 | 15.7 | **20.2** | 37.2 | 41.5 (+0.1) |
| RSD | 8B | 1B | 7B | $\delta = 0.7$ | 50.4 | 85.4 | 41.8 | **18.1** | 19.7 | 36.2 | 41.9 (+0.5) |
| RSD | 8B | 1B | 1.5B | $\delta^*$ | 50.0 | 84.1 | 43.4 | **18.1** | **20.2** | **38.8** | 42.4 (+1.0) |
| RSD | 8B | 1B | 7B | $\delta^*$ | 50.6 | **85.5** | 42.6 | **18.1** | 19.7 | 38.7 | **42.5 (+1.1)** |

**Datasets.** We evaluate our method on a diverse set of reasoning tasks, including GSM8K (Cobbe et al., 2021b), MATH500 (Hendrycks et al., 2021), MMLU STEM (Hendrycks et al., 2020), OlympiadBench (He et al., 2024), GaoKao-2023-En (Liao et al., 2024), GPQA (Rein et al., 2023), and Minerva Math (Lewkowycz et al., 2022).

**Baselines.** We consider three categories of baselines: (1) *Target model only:* This baseline uses the target model independently, requiring more cost than RSD. (2) *Draft model with or without PRM:* This category includes popular test-time scaling methods that achieve the best possible performance using the draft model. Specifically, we consider majority voting, Best-of-$N$ (BoN) (Brown et al., 2024; Cobbe et al., 2021a) that selects the highest-scoring response (last step) among $N$ candidates based on a PRM, beam search (Chen et al., 2024a) that leverages a PRM to choose the optimal decoding path, process Best-of-$N$ that samples $N$ candidate steps and selects the one with the high-

est reward. For majotity voting and Best-of-$N$, we prefer a large number of samplings (more cost than the target model only) to show their converged performance. (3) *Speculative decoding (SD):* We also include speculative decoding with a number of speculative tokens as 7, a method designed to accelerate inference (Leviathan et al., 2023).

**Default Setting.** All experiments were conducted on NVIDIA A100 GPUs, using vLLM (Kwon et al., 2023) as the backend. We use temperature = 0.7 and top_p = 0.8 for majority voting, (process) Best-of-$N$ and beam search, while setting temperature = 0 and top_p = 1 for the remaining methods. For process Best-of-$N$, beam search and RSD, we define a generation ended with \n\n as a reasoning step, and then apply a PRM to rate this step. We employ the binary step function (the second option in Table 1) as the weighting function and set $\delta = 0.7$. For brevity, RSD (7B/72B/7B) denotes RSD with a 7B draft model, a 72B target model and a 7B PRM. SD (7B/72B)

*Table 3.* Comparison with search-based methods. Beam Search and Process Best-of-$N$ use a 1.5B base model and a 1.5B PRM.

| Method | Setting | MATH 500 | GSM8K | Minerva Math |
|---|---|---|---|---|
| Single Model (1.5B) | - | 73.8 | 85.0 | 29.0 |
| Process Best-of-$N$ | $N = 8$ | 75.8 | 87.8 | 32.7 |
| Process Best-of-$N$ | $N = 16$ | 76.0 | 87.9 | 31.2 |
| Beam Search | beam size $= 4$ | 78.2 | 88.9 | 33.5 |
| Beam Search | beam size $= 8$ | 78.2 | 88.4 | 32.4 |
| RSD (1.5B/7B/1.5B) | $\delta$=0.7 | **82.6** | **94.5** | **34.6** |

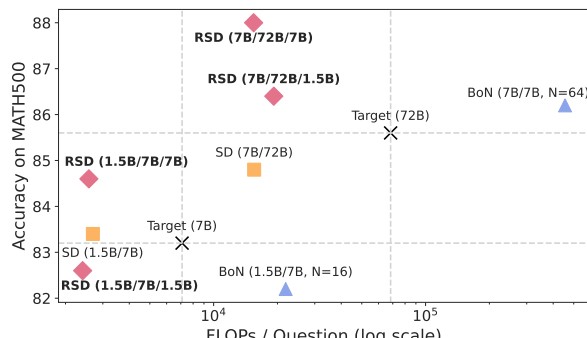

*Figure 4.* Flops vs. accuracy on MATH500.

deotes SD with a 7B draft model and a 72B target model. BoN (1.5B/1.5B) denotes BoN with a 1.5B base model and a 1.5B PRM. Without explicitly mentioning, models chosen from the Qwen-2.5-Math-Instruct family are used.

### 3.1. Reasoning Benchmarks

We evaluate RSD across a diverse set of reasoning benchmarks, as summarized in Table 2, and observe: (1) Test-time scaling methods like majority voting and Best-of-$N$, which rely on extensive sampling with a draft model, consistently underperform a single target model on average. This finding highlights the importance of a larger model for reasoning tasks, as its performance cannot be easily matched by a smaller model with increased computation. (2) While SD is theoretically unbiased, guaranteeing accuracy equal to the target model, it often underperforms in practice. This discrepancy, as also noted by Chen et al. (2023a), arises due to floating-point errors. Moreover, in cases where a draft model outperforms the target model (e.g., Table B.1 and domain-specialized draft models), SD's strict unbiasedness leads to worse performance compared to the draft model. Thus, the decision to use SD must account for such scenarios. In contrast, RSD mitigates this concern by leveraging a PRM, which evaluates the quality of reasoning steps from the draft model. (3) Among all evaluated methods, RSD consistently outperforms the single target model on average when using an optimized $\delta$. Even with a fixed $\delta = 0.7$, RSD achieves better results in 7 out of 8 settings. Notably, on the challenging GPQA benchmark, RSD (1.5B/7B/1.5B) significantly surpasses the single target model (38.4 vs. 32.8), demonstrating the effectiveness of this efficient approach.

Additionally, a larger PRM (7B) slightly enhances performance compared to a smaller PRM (1.5B), especially on complex datasets like GPQA and MATH500, where the increased reasoning capacity of a larger PRM proves beneficial. Results with general models, such as Qwen2.5-Instruct and Llama-3.1-Instruct, are even improved more, validating RSD's robustness and generalizability.

### 3.2. Comparison with Search-Based Methods

We also compare our method with beam search (Chen et al., 2024a) and process Best-of-$N$ in Table 3. RSD significantly

outperforms both search-based methods over all three benchmarks. These results highlight a critical insight: for certain complex or "hard" reasoning steps, search-based methods struggle to find optimal solutions due to the combinatorial explosion of potential candidates, leading to suboptimal performance. On the other hand, our approach leverages a larger model's capacity to generate plausible solutions more directly, bypassing the need for exhaustive search. By utilizing the PRM as a feedback mechanism, RSD benefits from step-wise guidance, which helps mitigate the challenges of reasoning in high-complexity tasks. This suggests that, rather than relying on a purely search-based strategy, incorporating larger models and targeted feedback mechanisms can lead to more efficient and effective reasoning, especially in cases where the search space is vast or the reasoning steps are particularly intricate.

### 3.3. Computation Analysis

To evaluate the computational efficiency of our method, we compare RSD with speculative decoding and Best-of-$N$ on MATH500. Following (Kang et al., 2024; Sardana et al., 2023), we adopt the standard approximation of FLOPs for transformers with $N$ parameters, i.e. $2N$ per inference token. Note that the inference cost for PRMs is also included in the calculations. As shown in Fig. 4, RSD (1.5B/7B/7B) outperforms both SD (1.5B/7B) and Target (7B), achieving an accuracy improvement of 1.2 and 1.4, respectively, while using fewer FLOPs. Moreover, RSD (7B/72B/7B) achieves a notable accuracy of 88.0 on MATH500, compared to 85.6 for Target (72B), with nearly $4.4\times$ fewer FLOPs. When compared to BoN (7B/7B,$N$=64), RSD (7B/72B/7B) delivers 1.8 points higher accuracy at a significantly lower computational cost. These results clearly demonstrate both efficiency and effectiveness of RSD.

### 3.4. Ablation Studies

**Threshold $\delta$.** Fig. 5 illustrates the relationship between the threshold $\delta$, accuracy, and the proportion of questions solved solely by the draft model within RSD. As $\delta$ increases, accuracy improves, peaking at $\delta = 0.7$, before experiencing

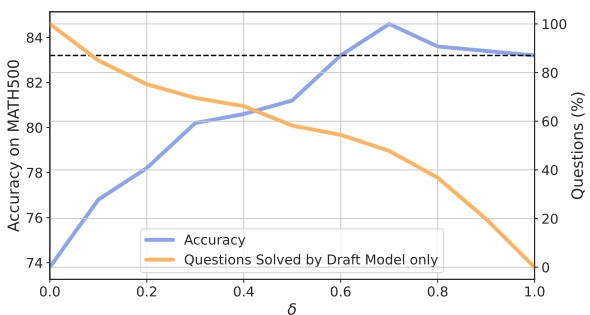

*Figure 5.* The impact of threshold $\delta$ with RSD (1.5B/7B/7B).

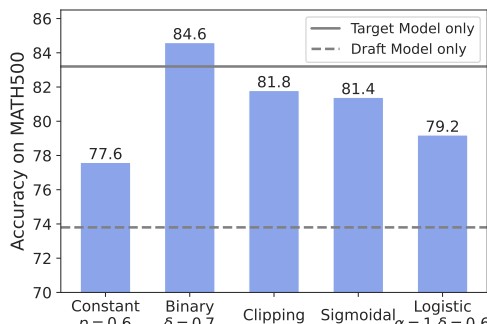

*Figure 6.* Accuracy of weighting functions from Table 1 with RSD (1.5B/7B/7B). All settings share a similar inference cost.

*Table 4.* Accuracy of model merge. * denotes that we merge these two models. Refer to §B.3 for detailed setting and number.

| Method | Target | Draft | PRM | Setting | Avg. Accuracy |
|---|---|---|---|---|---|
| Single Model | 7B | - | - | - | 65.3 |
| SD | 7B | 1.5B | - | - | 64.6 (-0.7) |
| RSD | 7B | 1.5B | 1.5B | $\delta = 0.7$ | 65.9 (+0.6) |
| RSD | 7B | 1.5B | 7B | $\delta = 0.7$ | 66.1 (+0.8) |
| RSD | 7B | 1.5B* | 1.5B* | $\delta = 0.7$ | 65.0 (-0.3) |
| RSD | 7B* | 1.5B | 7B* | $\delta = 0.7$ | **66.7 (+1.4)** |

a slight decline. Notably, the accuracy remains consistently higher than that of using only the target model ($\delta = 1.0$) once $\delta$ surpasses 0.6. A higher $\delta$ corresponds to a stricter rejection of reasoning steps generated by the draft model. However, even at $\delta = 0.7$, the draft model alone can still solve 48% of questions, as the rewards for its reasoning steps on these questions are sufficiently high. This eliminates the need for the target model to engage with these 48% of questions, distinguishing RSD from SD, which always involves both the draft and target models for every question.

This adaptability makes RSD an effective method for automatic compute allocation, allocating less compute (draft model only) to simpler questions and more compute (both draft and target models) to more challenging ones. For a more detailed discussion of the automatic compute allocation for questions in different levels, refer to §B.2. Additionally, $\delta$ serves as a critical role in balancing the computational efficiency of the draft model with the precision of the target model, ultimately optimizing overall performance.

**Weighting Function.** In Table 1, we present several candidate weighting functions. Here we evaluate their performance under comparable inference costs, as shown in Fig. 6. First, all candidates outperform the draft model alone, underscoring the critical role of incorporating a larger model within the reasoning loop. Second, the constant weighting function, which does not utilize rewards, performs the worst, emphasizing the significance of PRM feedback. Lastly, the binary step function achieves the best performance, even surpassing the single target model. Additionally, it introduces $\delta$ as a hyperparameter, allowing for flexible control over inference costs to accommodate varying budget constraints. Thus, we use the binary step function in this paper, and leave the exploration of other rejection schemes to future.

## 4. Discussion

**PRM Overheads and Model Merge.** In MATH500, the average number of reasoning steps per question is 18, suggesting that the PRM is invoked 18 times per question, akin to generating 18 tokens. Furthermore, even a tiny PRM (1.5B) outperforms the single target model in RSD accuracy

(see Table 2). Therefore, adding an additional PRM for RSD incurs minimal overhead compared to SD.

Here, we further investigate the possibility of merging models to enhance the usability of RSD by reducing the number of served models. As shown in Table 4, merging models does not necessarily degrade performance and remains superior to SD. Interestingly, merging larger models even results in performance improvements, consistent with observations reported by Yadav et al. (2024).

**Robustness to PRMs.** To investigate RSD's robustness to the choice of PRM, we include two more strong PRMs, i.e. Qwen2.5-Math-PRM-7B and 72B (Zhang et al., 2025) in Table 5. RSD equipped with different PRMs consistently outperforms SD, with more gains from a larger PRM.

**General-Domain Task.** An important component of RSD is PRM. To the best of our knowledge, there is not yet a PRM for general-domain generation. However, there are many outcome reward models (ORMs) for open-ended generation. Could we use an ORM instead of PRM in RSD?

Here, we utilize Llama-3.2-1B-Instruct (Grattafiori et al., 2024) as the draft model, Llama-3.1-8B-Instruct as the target model, and Skywork-Reward-Llama-3.1-8B-v0.2 (Liu et al., 2024) as the ORM. The 805 prompts from AlpacaEval (Dubois et al., 2023) are used for the generation. And the model outputs are evaluated with AlpacaEval2.0 (Li et al., 2023) against the outputs from gpt4_turbo. Similar to the setting for PRM, we define a generation ended with \n\n as a reasoning step, and apply the ORM to score this step. The score of Skywork-Reward-Llama-3.1-8B-v0.2 ranges from $-\infty$ to $\infty$. We didn't extensively tune the reward threshold,

*Table 5.* Accuracy on reasoning benchmarks with different PRMs. RSD is robust to the choice of PRM. $\delta = 0.7$ works well universally.

| Method | PRM | Setting | MATH500 | GSM8K | GaoKao 2023 En | Olympiad Bench | GPQA Diamond | MMLU STEM | Avg. |
|---|---|---|---|---|---|---|---|---|---|
| | | *Draft: Qwen2.5-Math-Instruct-1.5B, Target: Qwen2.5-Math-Instruct-7B* | | | | | | | |
| Single Target Model | - | - | 83.2 | 95.7 | 66.8 | 41.2 | 32.8 | 71.8 | 65.3 |
| SD | - | - | 83.4 | 95.6 | 67.3 | 40.6 | 28.8 | 72.0 | 64.6 (-0.7) |
| RSD | Skywork-o1-Open-PRM-1.5B | $\delta = 0.7$ | 82.6 | 94.5 | **68.8** | 39.6 | **38.4** | 71.4 | 65.9 (+0.6) |
| RSD | Skywork-o1-Open-PRM-7B | $\delta = 0.7$ | **84.6** | 95.5 | 68.3 | **42.1** | 33.8 | 72.3 | 66.1 (+0.8) |
| RSD | Qwen2.5-Math-PRM-7B | $\delta = 0.7$ | 83.2 | 95.5 | 67.3 | 41.3 | 35.3 | 72.5 | 65.9 (+0.6) |
| RSD | Qwen2.5-Math-PRM-72B | $\delta = 0.7$ | 84.0 | **95.7** | 68.1 | 42.0 | 33.3 | **75.6** | **66.5 (+1.2)** |

*Table 6.* Results on a general-domain benchmark, AlpacaEval. The PRM used here is an outcome reward model, Skywork-Reward-Llama-3.1-8B-v0.2.

| Method | Win Rate (%) |
|---|---|
| Draft Model only (Llama-3.2-1B-Instruct) | 7.09 |
| Target Model only (Llama-3.1-8B-Instruct) | 24.47 |
| RSD | 18.85 |

and empirically chose $\delta = 0$.

As shown in the In Table 6, even with an ORM instead of a PRM, RSD achieves a significantly better win rate than the draft model, showing RSD's robustness across different tasks. Among all generated tokens, 65% tokens are generated by the draft model only without any intervention of the target model. We believe that a general-domain PRM and dedicated tuning of $\delta$ could further boost the performance.

**Combine RSD with SD.** RSD is not inherently opposed to SD; in fact, they can be seamlessly combined to enhance efficiency. For instance, during a rejected step, SD (draft+target) can be utilized to regenerate the step. This approach allows for further optimization of RSD's efficiency without incurring additional costs.

**Specialized PRM.** In our experiments, we rely on an open-source general PRM. However, training or fine-tuning a specialized PRM that is closely aligned with the draft model could further enhance performance. Such a PRM would better recognize high-quality reasoning steps generated by the draft model, thereby increasing the acceptance rate. Future work could explore specialized PRM training or fine-tuning.

## 5. Related Work

**Speculative Decoding.** Speculative decoding (Stern et al., 2018; Leviathan et al., 2023; Xia et al., 2024; Chen et al., 2023a; Zhang et al., 2023a; Sun et al., 2024a; Chen et al., 2023b; Li et al., 2024b) achieves lossless acceleration by employing a draft model to predict subsequent tokens and verify them in parallel. Tree-based speculation (Miao et al., 2024; Fu et al., 2024; Sun et al., 2024b; Chen et al., 2024b) extends this approach by generating multiple candidates to increase the acceptance rate. Self-speculative decod-

ing (Elhoushi et al., 2024; Zhang et al., 2023a) leverages parts of the large language model (LLM) parameters as the draft model while using the original base model as the verifier. Parallel decoding (Stern et al., 2018; Cai et al., 2024) further enhances efficiency by introducing draft models to streamline the process. Unlike previous speculative decoding methods, our approach utilizes process rewards to perform stepwise speculative reasoning.

**Reward Models on Reasoning.** Reward models play a crucial role in selecting correct reasoning trajectories during training (Chen et al., 2024a; Wang et al., 2024; Zhou et al., 2025) and inference (Brown et al., 2024). Outcome Reward Models (ORMs) (Yu et al., 2023; Dong et al., 2024) are trained exclusively on the model's final output, whereas process reward models (PRMs) (Lightman et al., 2023) rely on step-level annotations, providing dense and granular reward signals at each reasoning step. Scaling test-time compute (Snell et al., 2024; OpenAI, 2024) has gained significant traction with the advancement of reward models. Techniques like Best-of-$N$ (Brown et al., 2024; Cobbe et al., 2021a; Dong et al., 2023) leverage ORMs to select the sample with the highest reward from N candidates. Building on this, tree search methods have been introduced, enabling per-step predictions rather than relying solely on the final answer (Chen et al., 2024a; Qi et al., 2024; Yao et al., 2024). These methods are enhanced by process feedback, such as process reward models (PRMs). We propose a novel application of PRMs to accelerate reasoning during inference.

## 6. Conclusion.

We propose Reward-Guided Speculative Decoding (RSD), a novel framework that enhances LLM inference efficiency, particularly for reasoning-intensive tasks. RSD dynamically combines a lightweight draft model with a more capable target model, using a reward function to guide output selection at each step. This approach balances computational cost and quality by selectively refining outputs based on process rewards. RSD achieves significant efficiency gains over SD and BoN while maintaining accuracy benchmarks. Extensive evaluations across reasoning tasks highlight RSD's robustness, adaptability, and effectiveness, making it a practical solution for LLM deployment.

## Acknowledgements

This research was partly supported by the Netherlands Organization for Scientific Research (NWO) under project number VI.C.192.080.

## Impact Statement

This paper presents work whose goal is to advance the field of Machine Learning by proposing Reward-Guided Speculative Decoding (RSD), a framework aimed at improving the efficiency and scalability of large language model inference. The potential societal implications of this work are primarily positive, as RSD facilitates more energy-efficient and cost-effective use of computational resources, contributing to the sustainability of deploying large-scale AI systems.

However, as with any advancement in machine learning, there are ethical considerations to be acknowledged. The increased efficiency of language models could lead to wider accessibility and adoption, which, while beneficial in many respects, may also exacerbate risks such as misuse for generating misinformation or biases in outputs. We strongly encourage researchers and practitioners to apply RSD responsibly and consider incorporating safeguards to mitigate these risks.

Overall, we believe the contributions of this work align with the broader goal of advancing AI technologies in an ethical and sustainable manner, with no immediate negative societal consequences requiring further discussion.

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

# A. Proof

## A.1. Proof of Proposition 2.1

*Proof.* For simplicity, we assume that $\tilde{y} = y|z$.

The final distribution $\mathbf{P}_{\mathrm{RSD}}(\tilde{y})$ combines contributions from two sampling paths:

- Accepted samples from $\mathbf{P}_m$.

- Fallback samples from $\mathbf{P}_M$ after rejection.

According to Law of Total Probability,

$$\mathbf{P}_{\mathrm{RSD}}(\tilde{y}) = \underbrace{\mathbf{P}(\text{accept } \tilde{y} \text{ from } \mathbf{P}_m)}_{\text{Term 1}} + \underbrace{\mathbf{P}(\text{reject } \mathbf{P}_m \text{ and draw } \tilde{y} \text{ from } \mathbf{P}_M)}_{\text{Term 2}}.$$

For accepted samples from $\mathbf{P}_m$, the acceptance probability for $\tilde{y} \sim \mathbf{P}_m$ is $\omega(r(\tilde{y}))$:

$$\text{Term 1} = \mathbf{P}_m(y) \cdot \omega(r(\tilde{y})).$$

For samples from $\mathbf{P}_M$, the rejection probability from $P_m$ is $\nu$, where:

$$\nu = 1 - \mathbf{E}_{\tilde{y} \sim P_m} [\omega(r(\tilde{y}))]$$

After rejection, $\tilde{y}$ is drawn from $\mathbf{P}_M$:

$$\text{Term 2} = \nu \cdot \mathbf{P}_M(y).$$

By combining two terms

$$\mathbf{P}_{\mathrm{RSD}}(\tilde{y}) = \mathbf{P}_m(y)\omega(r(\tilde{y})) + \nu\mathbf{P}_M(\tilde{y}),$$

where $\nu + \mathbf{E}_{\tilde{y} \sim P_m} [\omega(r(\tilde{y}))] = 1$.

$\square$

## A.2. Proof of Proposition 2.2

*Proof.* We aim to prove that under the conditions:

1. $\omega(r)$ is non-decreasing in $r$,

2. $\mathbf{E}_{\mathbf{P}_M}[r(y|z)] \geq \mathbf{E}_{\mathbf{P}_m}[r(y|z)]$,

the expectation of $r(y|z)$ under $\mathbf{P}_{\mathrm{RSD}}$ satisfies:

$$\mathbf{E}_{\mathbf{P}_{\mathrm{RSD}}}[r(y|z)] \geq \mathbf{E}_{\mathbf{P}_m}[r(y|z)].$$

By definition:

$$\mathbf{E}_{\mathbf{P}_{\mathrm{RSD}}}[r(y|z)] = \int \left(\omega(r(y|z))\mathbf{P}_m(y|z) + \nu\mathbf{P}_M(y|z)\right) r(y|z)dy$$

where $\nu = 1 - \mathbf{E}_{\mathbf{P}_m}[\omega(r(y|z))]$. Substituting $\nu$:

$$\mathbf{E}_{\mathbf{P}_{\mathrm{RSD}}}[r(y|z)] = \underbrace{\mathbf{E}_{\mathbf{P}_m}[\omega(r(y|z))r(y|z)]}_{\text{Term 1}} + \underbrace{(1 - \mathbf{E}_{\mathbf{P}_m}[\omega(r(y|z))])\mathbf{E}_{\mathbf{P}_M}[r(y|z)]}_{\text{Term 2}}.$$

The condition of $\mathbf{E}_{\mathbf{P}_{\mathrm{RSD}}}[r(y|z)] - \mathbf{E}_{\mathbf{P}_m}[r(y|z)] \geq 0$ can be rewritten as

$$\mathbf{E}_{\mathbf{P}_m}[\omega(r)r] + (1 - \mathbf{E}_{\mathbf{P}_m}[\omega(r)])\mathbf{E}_{\mathbf{P}_M}[r] - \mathbf{E}_{\mathbf{P}_m}[r] \geq 0.$$

Note that:

$$\mathbf{E}_{\mathbf{P}_m}[\omega(r)r] = \mathbf{Cov}_{\mathbf{P}_m}(\omega(r), r) + \mathbf{E}_{\mathbf{P}_m}[\omega(r)] \cdot \mathbf{E}_{\mathbf{P}_m}[r].$$

Substitute this into the inequality:

$$\mathbf{Cov}_{\mathbf{P}_m}(\omega(r), r) + \mathbf{E}_{\mathbf{P}_m}[\omega(r)]\mathbf{E}_{\mathbf{P}_m}[r] + (1 - \mathbf{E}_{\mathbf{P}_m}[\omega(r)])\mathbf{E}_{\mathbf{P}_M}[r] - \mathbf{E}_{\mathbf{P}_m}[r] \geq 0.$$

Simplify the terms involving $\mathbf{E}_{\mathbf{P}_m}[r]$:

$$\mathbf{Cov}_{\mathbf{P}_m}(\omega(r), r) + (1 - \mathbf{E}_{\mathbf{P}_m}[\omega(r)]) (\mathbf{E}_{\mathbf{P}_M}[r] - \mathbf{E}_{\mathbf{P}_m}[r]) \geq 0.$$

1. **Covariance Term ($\mathbf{Cov}_{\mathbf{P}_m}(\omega(r), r)$):** Since $\omega(r)$ is non-decreasing in $r$, higher values of $r$ correspond to higher values of $\omega(r)$. This implies $\mathbf{Cov}_{\mathbf{P}_m}(\omega(r), r) \geq 0$.

2. **Expectation Difference Term ($(1 - \mathbf{E}_{\mathbf{P}_m}[\omega(r)]) (\mathbf{E}_{\mathbf{P}_M}[r] - \mathbf{E}_{\mathbf{P}_m}[r])$):**
   - $1 - \mathbf{E}_{\mathbf{P}_m}[\omega(r)] = \nu \geq 0$ (since $\nu$ is a normalizing constant for a valid probability distribution).
   - By the second condition, $\mathbf{E}_{\mathbf{P}_M}[r] - \mathbf{E}_{\mathbf{P}_m}[r] \geq 0$.

   Thus, this term is non-negative.

Both terms in the inequality:

$$\mathbf{Cov}_{\mathbf{P}_m}(\omega(r), r) \geq 0 \quad \text{and} \quad (1 - \mathbf{E}_{\mathbf{P}_m}[\omega(r)]) (\mathbf{E}_{\mathbf{P}_M}[r] - \mathbf{E}_{\mathbf{P}_m}[r]) \geq 0$$

are non-negative under the given conditions. Therefore:

$$\mathbf{E}_{\mathbf{P}_{\mathrm{RSD}}}[r(y|z)] \geq \mathbf{E}_{\mathbf{P}_m}[r(y|z)].$$

The conditions $\omega(r)$ is non-decreasing in $r$ and $\mathbf{E}_{\mathbf{P}_M}[r(y|z)] \geq \mathbf{E}_{\mathbf{P}_m}[r(y|z)]$ are sufficient to guarantee $\mathbf{E}_{\mathbf{P}_{\mathrm{RSD}}}[r(y|z)] \geq \mathbf{E}_{\mathbf{P}_m}[r(y|z)]$. $\qquad\qquad\square$

### A.3. Proof of Proposition 2.3

*Proof.* For simplicity, we assume that $\tilde{y} = y|z$. Our optimization problem is

$$\mathcal{L}(\omega_r) = \mathbf{E}_{\tilde{y}\sim\mathbf{P}_m}\omega_r(\tilde{y})r(\tilde{y}) + \mathbf{E}_{\tilde{y}\sim\mathbf{P}_M}\nu r(\tilde{y})$$

subject to the inequality constraint:

$$\nu = 1 - \mathbf{E}_{\tilde{y}\sim\mathbf{P}_m}[\omega_r(\tilde{y})] \leq \gamma, \quad 0 \leq \omega(\tilde{y}) \leq 1.$$

Equivalently,

$$\mathcal{L}(\omega_r) = \mathbf{E}_{\mathbf{P}_m}[\omega_r(\tilde{y})\, r(\tilde{y})] + (1 - \mathbf{E}_{\mathbf{P}_M}[\omega_r(\tilde{y})]) \mathbf{E}_{\mathbf{P}_M}[r(\tilde{y})]$$

The Lagrangian[1] is given by

$$\mathcal{L}(\omega_r, \lambda) = \int \left[ (\mathbf{P}_m(\tilde{y})r(\tilde{y}))\omega_r(\tilde{y}) + (\mathbf{P}_M(\tilde{y})r(\tilde{y}))(1 - \mathbf{E}_{\tilde{y}}[\omega_r(\tilde{y})]) \right] d\tilde{y} + \lambda \left[ (1 - \gamma) - \int \mathbf{P}_m(\tilde{y})\omega_r(\tilde{y})d\tilde{y} \right].$$

The Lagrangian derivative yields:

$$\frac{\partial \mathcal{L}}{\partial \omega_r(\tilde{y})} = \mathbf{P}_m(\tilde{y}) \left[ r(\tilde{y}) - (\lambda + R) \right],$$

where $R = \mathbf{E}_{\tilde{y}\sim\mathbf{P}_M}r(\tilde{y})$.

Setting this to zero gives the threshold rule: the optimal $\omega_r^*(\tilde{y})$ is:

$$\omega_r^*(\tilde{y}) = \begin{cases} 1, & \text{if } r(\tilde{y}) - (\lambda + R) \geq 0, \\ 0, & \text{if } r(\tilde{y}) - (\lambda + R) < 0. \end{cases}$$

**KKT Conditions.**

---

[1] Here, the integral can also be defined for a counting measure space, allowing it to be applied to the discrete case.

- Primal feasibility: $\mathbf{E}_{\tilde{y}\sim\mathbf{P}_m}[\omega_r(\tilde{y})] \geq 1 - \gamma$.

- Dual feasibility: $\lambda \geq 0$.

- Stationarity: The first-order condition holds.

- Complementary slackness: $\lambda\left((1-\gamma) - \int \mathbf{P}_m(\tilde{y})\omega_r(\tilde{y})d\tilde{y}\right) = 0$.

**Conclusion.**

- If $\mathbf{E}_{\tilde{y}\sim\mathbf{P}_m}[\omega_r^*(\tilde{y})] = 1 - \gamma$, then $\lambda \geq 0$ and the solution is tight, as in the equality case.

- If $\mathbf{E}_{\tilde{y}\sim\mathbf{P}_m}[\omega_r^*(\tilde{y})] > 1 - \gamma$, then $\lambda = 0$, and the constraint is satisfied with slack.

- The solution is a threshold-based function on $r(\tilde{y})$, where $\omega_r^*(\tilde{y}) = 1$ for larger $r(\tilde{y})$ and $\omega_r^*(\tilde{y}) = 0$ for smaller $r(\tilde{y})$, adjusted to ensure that the constraint $\mathbf{E}_{\tilde{y}\sim\mathbf{P}_m}[\omega_r^*(\tilde{y})] \geq 1 - \gamma$ is satisfied.

Thus, The optimal sampling strategy involves "using $\mathbf{P}_m$" (i.e., $\omega_r(\tilde{y}) = 1$) for the higher values of $r(\tilde{y})$ and "using $\mathbf{P}_M$" (i.e., $\omega_r(\tilde{y}) = 0$) for the lower values of $r(\tilde{y})$. The threshold $t$ on $r(\tilde{y})$ is selected such that the constraint $\mathbf{E}_{\tilde{y}\sim\mathbf{P}_m}[\omega_r(\tilde{y})] \geq 1 - \gamma$ is met.

Threshold-based strategy: $\omega_r^*(\tilde{y}) = 1_{\{r(\tilde{y})\leq t\}}$, where $t$ is chosen to satisfy the constraint $\mathbf{E}_{\tilde{y}\sim\mathbf{P}_m}[\omega_r(\tilde{y})] \geq 1 - \gamma$.

$\square$

# B. Additional Empirical Results

*Table B.1.* Accuracy on CN Middle School 24 (Yang et al., 2024b) and College Math (Tang et al., 2024). Due to SD's strict unbiasedness, it achieves worse accuracy than the draft model if the draft model outperforms the target model, while RSD doesn't have this issue.

| Method | Target | Draft | PRM | Setting | CN Middle School 24 | College Math |
|---|---|---|---|---|---|---|
| Single Model | - | 1.5B | - | - | 75.2 | 48.0 |
| Single Model | 7B | - | - | - | 72.3 | 46.8 |
| SD | 7B | 1.5B | - | - | 73.3 | 46.9 |
| RSD | 7B | 1.5B | 7B | $\delta = 0.7$ | **78.2** | **48.2** |

## B.1. Tuning of $\delta$

RSD employs a threshold, $\delta$, to decide whether to accept a reasoning step generated by the draft model. If the reward score of a reasoning step exceeds $\delta$, it is accepted. However, reasoning tasks vary in complexity, leading to diverse reward distributions. Using a fixed $\delta$ may not yield optimal accuracy across different tasks.

The results for varying $\delta$ values are presented in Table B.2. Overall, $\delta = 0.7$ emerges as a reliable choice across various settings. Slight adjustments within the range [0.6, 0.7, 0.8, 0.9] can further improve performance.

## B.2. Different Reasoning Complexity

Thanks to the human annotated complexity levels in MATH500 (5 levels, the higher the harder), here we investigate how RSD works for questions in different complexity. As shown in Fig. B.1, the involvement of the target model ($\delta \neq 0$) consistently improves the accuracy compared with the draft model only ($\delta = 0$). The improvement varies for different levels, at most +4.7 for level 1, +5.6 for level 2, +6.7 for level 3, +16.4 for level 4 and +15.7 for level 5, showing the importance of the target model for harder questions.

For the same $\delta$, one can also observe that the proportion of questions solved by the draft model alone decreases with an increasing level. For example at $\delta = 0.7$, draft model alone solves 84% questions in level 1, 67% questions in level 2, 58% questions in level 3, 44% questions in level 4 and 19% questions in level 5. It shows that harder questions need more involvement of the target model. In this way, RSD can be considered as a method for automatic compute allocation, less compute for easy questions and more compute for hard questions, which is different from SD that always needs both target and draft models for every question.

*Table B.2.* Accuracy with different $\delta$s. Overall, $\delta = 0.7$ works well for different models and tasks. However, since the complexity of different tasks varies, a slight tuning of $\delta$ offers better accuracy.

| Method | Target Model | Draft Model | PRM | Setting | MATH500 | GSM8K | GaoKao 2023 En | Olympiad Bench | GPQA Diamond | MMLU STEM | Avg. |
|---|---|---|---|---|---|---|---|---|---|---|---|
| \multicolumn | | | | | | | | | | | |

*Math Model, Target and Draft: Qwen2.5-Math-Instruct, PRM: Skywork- o1-Open-PRM*

| Method | Target Model | Draft Model | PRM | Setting | MATH500 | GSM8K | GaoKao 2023 En | Olympiad Bench | GPQA Diamond | MMLU STEM | Avg. |
|---|---|---|---|---|---|---|---|---|---|---|---|
| RSD | 7B | 1.5B | 1.5B | $\delta = 0.6$ | 80.4 | 93.3 | 67.8 | **40.6** | 32.3 | 69.1 | 63.9 |
| RSD | 7B | 1.5B | 1.5B | $\delta = 0.7$ | **82.6** | 94.5 | **68.8** | 39.6 | **38.4** | 71.4 | **65.9** |
| RSD | 7B | 1.5B | 1.5B | $\delta = 0.8$ | **82.6** | 95.3 | 68.1 | 39.4 | 37.4 | 71.7 | 65.8 |
| RSD | 7B | 1.5B | 1.5B | $\delta = 0.9$ | 81.2 | **95.5** | 68.3 | 39.4 | 32.3 | **71.6** | 64.7 |
| RSD | 7B | 1.5B | 7B | $\delta = 0.6$ | 83.6 | 95.4 | **68.3** | 43.6 | 30.3 | 71.6 | 65.5 |
| RSD | 7B | 1.5B | 7B | $\delta = 0.7$ | **84.6** | 95.5 | **68.3** | 42.1 | 33.8 | 72.3 | **66.1** |
| RSD | 7B | 1.5B | 7B | $\delta = 0.8$ | 83.6 | **95.8** | 67.8 | 40.9 | **34.3** | **72.6** | 65.8 |
| RSD | 7B | 1.5B | 7B | $\delta = 0.9$ | 83.4 | 95.7 | 68.1 | 40.1 | 32.8 | 72.5 | 65.4 |
| RSD | 72B | 1.5B | 1.5B | $\delta = 0.6$ | 83.0 | 93.5 | 71.9 | **48.4** | 36.9 | 78.1 | 68.6 |
| RSD | 72B | 1.5B | 1.5B | $\delta = 0.7$ | 83.6 | 94.7 | **72.7** | 47.7 | 40.4 | 82.5 | 70.3 |
| RSD | 72B | 1.5B | 1.5B | $\delta = 0.8$ | **86.6** | **95.6** | 71.4 | 48.0 | 40.9 | 85.1 | 71.3 |
| RSD | 72B | 1.5B | 1.5B | $\delta = 0.9$ | 85.4 | **95.6** | 72.2 | 47.7 | **44.4** | **85.5** | **71.8** |
| RSD | 72B | 1.5B | 7B | $\delta = 0.6$ | 85.8 | 96.0 | 72.5 | **49.0** | 41.9 | 82.5 | 71.3 |
| RSD | 72B | 1.5B | 7B | $\delta = 0.7$ | 86.8 | **96.3** | 72.7 | **49.0** | 41.9 | 84.6 | **71.9** |
| RSD | 72B | 1.5B | 7B | $\delta = 0.8$ | **87.4** | **96.3** | 73.0 | 48.3 | 40.9 | 85.4 | **71.9** |
| RSD | 72B | 1.5B | 7B | $\delta = 0.9$ | 86.2 | 96.0 | 72.7 | 48.9 | 41.4 | **85.6** | 71.8 |
| RSD | 72B | 7B | 1.5B | $\delta = 0.6$ | 85.0 | 97.0 | 72.2 | 48.4 | **43.9** | 82.2 | 71.5 |
| RSD | 72B | 7B | 1.5B | $\delta = 0.7$ | 86.4 | 96.4 | 73.0 | **49.8** | 42.9 | 84.7 | 72.2 |
| RSD | 72B | 7B | 1.5B | $\delta = 0.8$ | **86.6** | 96.6 | 73.2 | 48.7 | **43.9** | **85.2** | **72.4** |
| RSD | 72B | 7B | 1.5B | $\delta = 0.9$ | 86.4 | 95.7 | 71.4 | 48.1 | 41.4 | 85.1 | 71.4 |
| RSD | 72B | 7B | 7B | $\delta = 0.6$ | **88.0** | 96.6 | 72.5 | 49.6 | 40.4 | 82.5 | 71.6 |
| RSD | 72B | 7B | 7B | $\delta = 0.7$ | **88.0** | 96.7 | **74.0** | 49.9 | 42.9 | 84.4 | **72.7** |
| RSD | 72B | 7B | 7B | $\delta = 0.8$ | 87.4 | **96.9** | 73.5 | 48.3 | 41.9 | **85.6** | 72.3 |
| RSD | 72B | 7B | 7B | $\delta = 0.9$ | 86.0 | 96.2 | 73.5 | 48.3 | 42.4 | 85.4 | 72.0 |
| \multicolumn | | | | | | | | | | | |

*General Model, Target and Draft: Qwen2.5-Instruct, PRM: Skywork- o1-Open-PRM*

| Method | Target Model | Draft Model | PRM | Setting | MATH500 | GSM8K | GaoKao 2023 En | Olympiad Bench | GPQA Diamond | MMLU STEM | Avg. |
|---|---|---|---|---|---|---|---|---|---|---|---|
| RSD | 7B | 1.5B | 1.5B | $\delta = 0.6$ | 72.8 | 89.7 | 63.6 | 38.5 | 22.2 | **71.7** | 59.8 |
| RSD | 7B | 1.5B | 1.5B | $\delta = 0.7$ | 73.6 | 90.8 | 64.2 | 39.0 | **31.3** | 71.6 | **61.8** |
| RSD | 7B | 1.5B | 1.5B | $\delta = 0.8$ | **74.8** | 91.6 | 64.4 | 38.8 | 23.7 | 67.1 | 60.1 |
| RSD | 7B | 1.5B | 1.5B | $\delta = 0.9$ | 74.6 | **92.3** | **65.2** | **40.0** | 28.3 | 59.3 | 60.0 |
| RSD | 7B | 1.5B | 7B | $\delta = 0.6$ | 75.4 | 92.6 | 64.9 | 37.3 | **29.8** | **66.3** | 61.1 |
| RSD | 7B | 1.5B | 7B | $\delta = 0.7$ | 75.0 | **93.3** | **66.2** | 39.9 | 20.2 | 61.8 | 59.4 |
| RSD | 7B | 1.5B | 7B | $\delta = 0.8$ | 74.2 | 92.4 | 62.9 | 40.6 | 21.7 | 58.2 | 58.3 |
| RSD | 7B | 1.5B | 7B | $\delta = 0.9$ | **75.8** | 92.1 | 65.2 | **40.9** | 26.8 | 56.1 | 59.5 |
| \multicolumn | | | | | | | | | | | |

*General Model, Target and Draft: Llama-3.1-Instruct, PRM: Skywork- o1-Open-PRM*

| Method | Target Model | Draft Model | PRM | Setting | MATH500 | GSM8K | GaoKao 2023 En | Olympiad Bench | GPQA Diamond | MMLU STEM | Avg. |
|---|---|---|---|---|---|---|---|---|---|---|---|
| RSD | 8B | 1B | 1.5B | $\delta = 0.6$ | 49.0 | 82.6 | 40.8 | **18.1** | 19.2 | 34.5 | 40.7 |
| RSD | 8B | 1B | 1.5B | $\delta = 0.7$ | **50.0** | 83.9 | 41.8 | 15.7 | **20.2** | 37.2 | **41.5** |
| RSD | 8B | 1B | 1.5B | $\delta = 0.8$ | 48.6 | **84.1** | 41.8 | 16.3 | 18.7 | **38.8** | 41.4 |
| RSD | 8B | 1B | 1.5B | $\delta = 0.9$ | **50.0** | 84.0 | **43.4** | 15.3 | 17.7 | 38.6 | **41.5** |
| RSD | 8B | 1B | 7B | $\delta = 0.6$ | 50.4 | **85.5** | 42.6 | 16.9 | 18.2 | 34.9 | 41.4 |
| RSD | 8B | 1B | 7B | $\delta = 0.7$ | 50.4 | 85.4 | 41.8 | **18.1** | **19.7** | 36.2 | **41.9** |
| RSD | 8B | 1B | 7B | $\delta = 0.8$ | **50.6** | 84.2 | 41.3 | 16.4 | 18.2 | 37.9 | 41.4 |
| RSD | 8B | 1B | 7B | $\delta = 0.9$ | 50.0 | 83.5 | 42.3 | 16.1 | 18.2 | **38.7** | 41.5 |

*Table B.3.* Accuracy of model merge. $^*$ denotes that we merge these two models.

| Method | Target Model | Draft Model | PRM | Setting | MATH500 | GSM8K | GaoKao 2023 En | Olympiad Bench | GPQA Diamond | MMLU STEM | Avg. |
|---|---|---|---|---|---|---|---|---|---|---|---|
| \multicolumn | | | | | | | | | | | |

*Math Model, Target and Draft: Qwen2.5-Math-Instruct, PRM: Skywork- o1-Open-PRM*

| Method | Target Model | Draft Model | PRM | Setting | MATH500 | GSM8K | GaoKao 2023 En | Olympiad Bench | GPQA Diamond | MMLU STEM | Avg. |
|---|---|---|---|---|---|---|---|---|---|---|---|
| Single Model | 7B | - | - | - | 83.2 | 95.7 | 66.8 | 41.2 | 32.8 | 71.8 | 65.3 |
| SD | 7B | 1.5B | - | - | 83.4 | **95.6** | 67.3 | 40.6 | 28.8 | 72.0 | 64.6 (-0.7) |
| RSD | 7B | 1.5B | 1.5B | $\delta = 0.7$ | 82.6 | 94.5 | 68.8 | 39.6 | **38.4** | 71.4 | 65.9 (+0.6) |
| RSD | 7B | 1.5B | 7B | $\delta = 0.7$ | **84.6** | 95.5 | 68.3 | 42.1 | 33.8 | 72.3 | 66.1 (+0.8) |
| RSD | 7B | 1.5B$^*$ | 1.5B$^*$ | $\delta = 0.7$ | 83.4 | 95.1 | 67.3 | 39.3 | 33.3 | 71.6 | 65.0 (-0.3) |
| RSD | 7B$^*$ | 1.5B | 7B$^*$ | $\delta = 0.7$ | 84.0 | **95.6** | **69.4** | **43.0** | 35.4 | **72.6** | **66.7 (+1.4)** |

## B.3. Model Merge

To reduce the number of models required for facilitating RSD's usage, we consider merging either the target model with the PRM or the draft model with the PRM. Here, we focus on the simplest merging strategy—linear merging—using

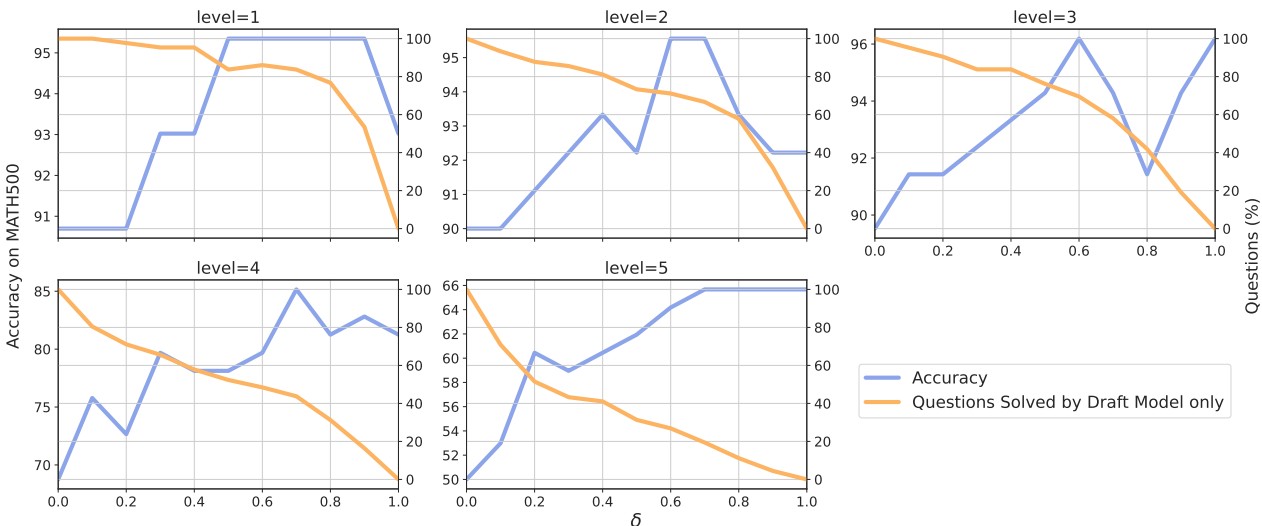

*Figure B.1.* The behaviour of RSD (1.5B/7B/7B) for different $\delta$s and questions in different complexity levels (the higher the level, the harder the question.). $\delta = 0$ and $\delta = 1$ denotes all questions are solved by the draft model alone and the target model only, respectively. Overall, the involvement of the target model improves the accuracy. The improvement is more obvious for harder question, +16 for level 4 and 5. In addition, with an increasing level, the questions solved by the draft model only decrease for the same $\delta$, demonstrating harder questions need more involvement of the target model.

MergeKit (Goddard et al., 2024), leaving the exploration of more advanced merging methods for future work.

The PRM and the policy model have different architectures. Specifically, the PRM includes a projection layer atop the final transformer layer (Vaswani et al., 2017), which projects the hidden dimension to a scalar output, whereas the policy model employs an `lm_head`. We merge only the shared layers, retaining the PRM's projection layer and the policy model's `lm_head`. For interpolation weights in the linear merging process, we tested only [0.6, 0.4] and [0.5, 0.5], with the target or draft model receiving 0.6 and the PRM 0.4. The [0.6, 0.4] configuration performed slightly better.

As shown in Table B.3, the results indicate the following: (1) Overall, the merged model outperforms SD; (2) Merging improves performance more substantially in larger models (+1.4 vs. +0.8). This observation aligns with the findings of Yadav et al. (2024).

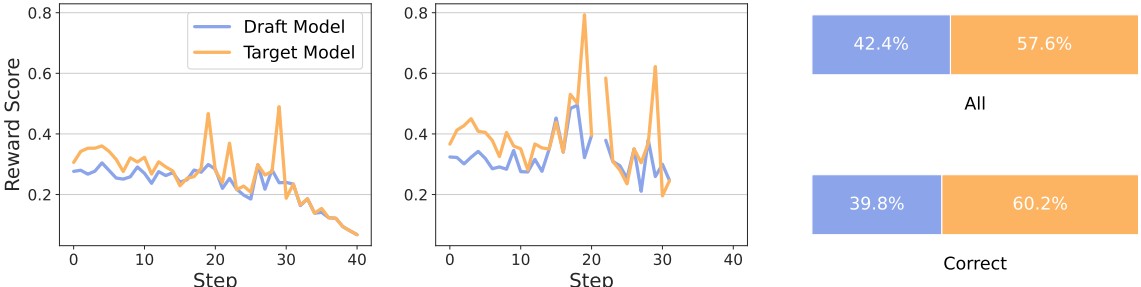

*Figure B.2.* **Left**: A comparison of the reward scores for all questions generated by the draft model and the target model within the RSD framework. **Middle**: A focused comparison of the reward scores for correctly answered questions generated by the draft model and the target model in the RSD framework. **Right**: The winning rate comparison between the draft model and the target model, highlighting the proportion of cases where each model outperforms the other in the RSD framework. RSD is configured with Qwen2.5-Math-1.5B-Instruct as the draft model, Qwen2.5-Math-72B-Instruct as the target model, and Skywork-o1-Open-PRM-7B as the PRM.

