# OpenReview forum: "Reward-Guided Speculative Decoding for Efficient LLM Reasoning"
_ICML.cc/2025/Conference — ICML 2025 poster_

### Official Review · Reviewer_XUtc · 2025-03-14

**Overall Recommendation:** 3

**Summary:**

The paper introduces Reward-Guided Speculative Decoding (RSD), an improved version of SD which replies on a process reward model to determine the quality of a step instead of exact match. RSD combines a lightweight draft model with a more capable target model, integrating a reward-based mechanism to optimize computational cost and output quality. RSD allows controlled bias by accepting high-reward outputs from the draft model, even when they do not perfectly match the target model’s predictions. The authors provide theoretical justification for a threshold-based mixture strategy, ensuring an optimal balance between computational efficiency and performance. Empirical results on various reasoning benchmarks, including GSM8K, MATH500, OlympiadBench, GPQA, MMLU STEM, and GaoKao-2023-En, demonstrate that RSD achieves up to 4.4× fewer FLOPs while improving accuracy by up to +3.5 points compared to standard speculative decoding.

**Claims And Evidence:**

Most claims in the paper are well-supported with both theoretical analysis and empirical evidence:
1. The claim that RSD reduces computational cost while maintaining or improving accuracy is supported through detailed FLOPs analysis and comparisons across multiple benchmarks.
2. The claim that RSD outperforms traditional speculative decoding (SD) on reasoning tasks is substantiated with controlled experiments using various LLM sizes and evaluation datasets.

Some aspects could benefit from additional justification:
1. The assertion that RSD can “even surpass the large model’s performance” requires more scrutiny. While some results suggest that RSD achieves higher accuracy than the single target model, this may depend on careful hyperparameter tuning (e.g., choice of threshold δ), and the exact conditions under which this occurs need further clarification.
2. The theoretical analysis of reward-guided acceptance could be extended by considering alternative reward models or potential biases introduced by specific reward function choices.

Overall, the claims are convincing, but further ablation studies on reward function variations and distribution mismatch effects would strengthen the argument.

**Essential References Not Discussed:**

No

**Experimental Designs Or Analyses:**

I reviewed the experimental setup, and it appears to be methodologically sound:

1. Controlled comparisons with speculative decoding and single-model baselines ensure a fair evaluation.
2. Hyperparameter tuning for δ is systematically explored, showing that a threshold around δ = 0.7 balances efficiency and accuracy.
3. Computational cost is measured in FLOPs, providing an objective comparison.

Limits:
1. One possible limitation is that the experiments primarily focus on math and reasoning tasks. It would be interesting to see how RSD performs on open-ended generation tasks (e.g., summarization, dialogue).
2. The choice of process reward model (PRM) is not extensively analyzed—would a different PRM significantly change RSD’s effectiveness?

Overall, the experimental setup is robust, but future work could explore how different PRMs influence performance and whether RSD generalizes beyond reasoning tasks.

**Methods And Evaluation Criteria:**

Yes, the proposed methods and evaluation criteria are well-aligned with the problem of efficient inference for LLM reasoning:

1. Evaluation on a broad set of reasoning benchmarks (GSM8K, MATH500, OlympiadBench, etc.) is appropriate because speculative decoding has traditionally been underutilized for multi-step reasoning tasks.
2. Comparisons with baseline methods (single target model, speculative decoding, Best-of-N, beam search) ensure a fair assessment of RSD's performance.
3. Computational efficiency is rigorously analyzed using FLOPs per question, which is a relevant metric for assessing inference cost.

However, a more detailed breakdown of performance by question difficulty (e.g., comparing RSD with SD for simple vs. complex problems) could provide deeper insights into when RSD is most beneficial.

The chosen evaluation methods are sound, but the paper could further explore RSD’s robustness to different reward models and variations in reasoning complexity.

**Other Comments Or Suggestions:**

The authors discussed about the choice of \delta and some experiments to explore the pick of it. I am wondering if it is possible to make the selection of the parameter as dynamic based on the difficulty of the question.

**Other Strengths And Weaknesses:**

Weakness:
The performance of the PRM is quite important for the process, yet this is can also be the limit of performance. Although the result shows it can go beyond the target model, but still limited.

**Questions For Authors:**

No.

**Relation To Broader Scientific Literature:**

The paper builds on prior work in speculative decoding and efficient inference for LLMs, making several novel contributions:

1. Improves upon speculative decoding (SD): Prior SD methods (Leviathan et al., 2023) enforce strict unbiasedness, whereas RSD introduces reward-guided acceptance to enhance efficiency.
2. Incorporates a process reward model (PRM): Similar to reinforcement learning approaches (Dong et al., 2023), but applied to speculative decoding for stepwise validation of reasoning.
3. Demonstrates efficiency gains over standard decoding strategies: The results align with recent efforts to reduce inference cost via speculative sampling (Chen et al., 2023) and model acceleration (Frantar et al., 2022).

**Theoretical Claims:**

The proofs in the paper are logically consistent, they assume an idealized reward function without considering potential noise in reward estimation. A discussion on the sensitivity of these theoretical results to imperfect reward models would be useful.

---

> ### Author Rebuttal · Authors · 2025-04-01
>
> $~$
>
> **Supplementary Material:** https://anonymous.4open.science/r/Rebuttal-supp-6595-C4A2/appendix_rebuttal_ICML.pdf
>
> $~$
>
> ---
>
> **Q1.** Why RSD outperforms the large model?
>
> > Please refer to our rebuttal to **Reviewer p3vx Q5**.
>
> $~$
>
> **Q2.** Alternative reward models or potential biases
>
> > As we mentioned, one might consider alternative formulations:
> >- **Likelihood-Ratio-Based:** One could incorporate a term like $\frac{\mathbf{P}_M(y|z)}{\mathbf{P}_m(y|z)}$ to define the weighting function (e.g., $w(y|z)=\min(1, \alpha\,\frac{\mathbf{P}_M(y|z)}{\mathbf{P}_m(y|z)})$). Such a formulation may capture discrepancies between the two models’ distributions, but it may also introduce bias if the likelihood estimates are themselves noisy or miscalibrated.
> >- **Hybrid:** One might combine the process reward and likelihood ratio—e.g., $w(y|z)=\min(1, \beta\,r(y|z)\,\frac{\mathbf{P}_M(y|z)}{\mathbf{P}_m(y|z)})$. While this hybrid approach can potentially leverage complementary strengths, its sensitivity to misestimations in either component may induce systematic biases. For example, if the reward function overestimates quality in regions where the draft model already performs well, the weighting may overfavor the draft model, reducing the benefits of the target model’s corrections.
>
> >In both cases, under our assumption, the underlying distribution will be better than small model induced one. Whereas, the choice of reward function can significantly affect the acceptance probability, and thus the final mixture distribution. When we choose more weights towards likelihood, it will biased to exact-match of large model distribution, so that the cost would be higher (similar to SD).
>
> >You may also consider imperfect reward (Q5).
>
> $~$
>
> **Q3.** Performance by question difficulty.
>
> > We show it in Tab C.4 (in link). One can observe:
> > * For simpler questions (level=1), RSD and SD performs the same;
> > * For harder questions (level>2), RSD consistently outperforms SD.
> > With the help of PRM, RSD selectively includes the target model for some reasoning steps, correcting the wrong reasoning step and leading to better performance.
>
> $~$
>
> **Q4.** Different reward models & variations in reasoning complexity.
>
> > In Tab C.5 (in link), we includes more PRMs, Qwen2.5-Math-PRM-7B and Qwen2.5-Math-PRM-72B, released during the reviewing cycle.
> > * RSD is robust to PRM. Although Skywork-o1-Open-PRM (regression model) and Qwen2.5-Math-PRM (classification model with num_labels=2) are trained differently, they all perform quite well across different tasks.
> > * $\delta=0.7$ consistently yields strong results across all four PRMs, appearing to be a sweet spot for balancing efficiency and performance when the reward score lies in [0, 1].
>
> > We also draw the accuracy and efficiency wrt $\delta$ in Fig C.3 (in link). These two PRMs behave similarly:
> > * Involving the target model ($\delta \neq 0$) consistently improves accuracy over using the draft model alone ($\delta = 0$), with greater gains on harder questions.
> > * For the same $\delta$, the proportion of questions solved by the draft model alone decreases with an increasing level, showing that harder questions need more involvement of the target model.
>
> $~$
>
> **Q5.** Imperfect reward models
>
> > In practice, $\hat{r}(y|z)$ is an approximation of the oracle reward $r_{\text{oracle}}(y|z)$. We can derive non-asymptotic convergence bounds under assumptions on the reward model’s error distribution.
>
> > *Assumption (Sub-Gaussian Error):* Assume that for each reasoning step, the estimation error
> $$
> \epsilon(y|z) = \hat{r}(y|z) - r_{\text{oracle}}(y|z)
> $$
> is sub-Gaussian with parameter $\sigma^2$; that is, for any \(t > 0\),
> $$
> \Pr\left(|\epsilon(y|z)| \geq t\right) \leq 2\exp\left(-\frac{t^2}{2\sigma^2}\right).
> $$
> > Under this assumption, when using $n$ independent observations, standard concentration inequalities imply that, with probability at least $1-\delta$,
> $$
> \left|\frac{1}{n}\sum_{i=1}^n \hat{r}(y_i|z) - \mathbf{E}[r_{\text{oracle}}(y|z)]\right| \leq \sigma \sqrt{\frac{2\log(2/\delta)}{n}}.
> $$
>
> > Even if the reward model is imperfect, the empirical average reward converges to the oracle at $O\Bigl(\sqrt{\frac{\log(1/\delta)}{n}}\Bigr)$.
> The bias introduced in the acceptance decision and in  $\mathbf{P}_{\text{RSD}}$ can be controlled in a non-asymptotic manner. Even with imperfect reward estimates, the impact on the RSD  diminishes as more data become available.
>
> $~$
>
> **Q6.** Open-ended generation tasks.
>
> > Refer to Q2 of Reviewer GXeJ.
>
> $~$
>
> **Q7.** Choice of $\delta$ based on the difficulty.
>
> > From Fig C.1, $\delta=0.7$ performs consistently well across difficulty levels, achieving top accuracy in 4/5 levels. Q4 further confirms $\delta=0.7$ works well for different PRMs.
> >
> > That said, dynamically selecting $\delta$ based on question difficulty (eg using a LLM to predict difficulty and adjust $\delta$ accordingly) could improve performance and efficiency. We leave this for future work.

---

### Official Review · Reviewer_GXeJ · 2025-03-14

**Overall Recommendation:** 4

**Summary:**

This paper introduces Reward-Guided Speculative Decoding (RSD), a novel framework designed to improve the efficiency of inference in large language models (LLMs) by combining a lightweight draft model with a more powerful target model. Extensive evaluations on challenging reasoning benchmarks demonstrate that RSD significantly reduces computational costs (up to 4.4× fewer FLOPs) while improving accuracy compared to using the target model alone or parallel decoding methods. The results highlight RSD as a robust and cost-effective approach for deploying LLMs in resource-intensive scenarios, particularly for complex reasoning tasks.

**Claims And Evidence:**

The paper claims that Reward-Guided Speculative Decoding (RSD) improves LLM inference efficiency by combining a draft model with a target model, guided by a process reward model. Evidence includes up to 4.4× fewer FLOPs and improved accuracy on reasoning benchmarks like MATH500 and GSM8K. The results show RSD outperforms traditional methods and models.

**Essential References Not Discussed:**

N/A

**Experimental Designs Or Analyses:**

Experiments compare RSD to baseline methods like speculative decoding and target-only models across multiple reasoning benchmarks. Metrics include FLOPs, accuracy, and the proportion of questions solved by the draft model alone. Ablation studies explore the impact of threshold values and weighting functions.

**Methods And Evaluation Criteria:**

RSD uses a draft model to generate candidate outputs, which are evaluated by a reward model to decide whether to invoke the target model. Evaluation is based on computational efficiency (FLOPs) and accuracy across reasoning tasks. Benchmarks include GSM8K, MATH500, and Olympiad-level tasks.

**Other Comments Or Suggestions:**

How does RSD perform on non-reasoning tasks like text generation or summarization?

**Other Strengths And Weaknesses:**

A weakness is the reliance on a process reward model, which may introduce additional overhead.

**Questions For Authors:**

Could the framework be extended to handle multimodal inputs?

**Relation To Broader Scientific Literature:**

RSD builds on speculative decoding and process reward models, addressing limitations in traditional methods. It aligns with research on efficient LLM inference and reasoning tasks.

**Theoretical Claims:**

The paper claims that a threshold-based mixture strategy optimally balances computational cost and output quality. Theoretical analysis shows that RSD ensures higher expected rewards compared to using the draft model alone.

---

> ### Author Rebuttal · Authors · 2025-04-01
>
> **Q1.** The reliance on a process reward model, which may introduce additional overhead.
>
> > This is indeed a very practical concern, since RSD utilizes one more model, i.e. the process reward model (PRM), than speculative decoding (SD). However, accoding to our experiments, the overhead is minor from three perspectives:
> > * **The PRM is small:** According to Table 2, a 1.5B PRM could already offers consistent better performance than all baselines with different draft and target models.
> > * **The inference cost of PRM is minimal:** RSD applies PRM to score the step instead of tokens from the draft model. The average number of steps per question in MATH500 is 18. The inference cost of PRM is similar to generate 18 tokens per question.
> > * **The PRM can be merged with the target or draft model:** In Table 4, we explore the possibility of merging the PRM with the draft or target model. Surprisingly, model merging doesn't obviously degrade the performance, and even results in better performance when merging the larger models. In this way, RSD shares the same number of models as SD. A further investigation of model merging between PRM and proxy model is left to future work.
>
> $~$
>
> **Q2.** How does RSD perform on non-reasoning tasks like text generation or summarization?
>
> > This is a very interesting suggestion. An important component of RSD is PRM. As far as we know, there is not yet a PRM for general-domain generation. But there are plenty of outcome reward models (ORMs) for open-ended generation. **Could we use ORM instead of PRM in RSD?**
>
> > **Experimental setup:**
> We utilize Llama-3.2-1B-Instruct as the draft model, Llama-3.1-8B-Instruct as the target model, and Skywork-Reward-Llama-3.1-8B-v0.2 as the ORM. The 805 prompts from AlpacaEval [1] are used for the generation, and the model outputs are evaluated with AlpacaEval2.0 against the outputs from gpt4_turbo. Similar to the setting in the paper, we define a generation ended with "\n\n" as a reasoning step, and apply the ORM to score this step. The score of Skywork-Reward-Llama-3.1-8B-v0.2 ranges from -$\infty$ to $\infty$. We didn't extensively tune the reward threshold $\delta$, and empirically chose $\delta=0$.
>
> > **Results:**
> As shown in the following table, even with an ORM instead of a PRM, RSD achieves a significantly better win rate than the draft model, showing RSD's robustness across different tasks. Among all generated tokens, 65% tokens are generated by the draft model only without any intervention of the target model. We believe that a general-domain PRM and dedicated tuning of $\delta$ could further boost the performance.
>
> | Method | Win Rate (%) against gpt4_turbo |
> | --- | --- |
> | Single Draft Model | 7.09 |
> | Single Target Model | 24.47 |
> | RSD | 18.85 |
>
> $~$
>
> **Q3.** Could the framework be extended to handle multimodal inputs?
>
>  > Yes, we believe our RSD could be seamlessly extended to the multomodal reasoning tasks, because:
>  > * The key component of RSD is PRM. As shown in Q2, even an ORM could act as a PRM. We believe that the existing multimodal ORM can be used in RSD for multimodal reasoning tasks.
>  > * We noticed that a new multimodal PRM [2] is released recently. It can be used perfectly in our framework.
> >
> > However, the exploration of multimodal reasoning tasks is out of the scope of this work, and there isn't any multimodal PRM when we conducted this research, we leave this investigation to interested readers.
>
> $~$
>
>  > [1] Length-Controlled AlpacaEval: A Simple Way to Debias Automatic Evaluators. Yann Dubois, Balázs Galambosi, Percy Liang, Tatsunori B. Hashimoto
>  >
>  >[2] VisualPRM: An Effective Process Reward Model for Multimodal Reasoning. Weiyun Wang, Zhangwei Gao, et. al., Wenhai Wang
>
> $~$
>
> ---
>
> Thank you very much for your thoughtful suggestions, making our work more solid. We have incorporated the new results in the updated version by our side.
>
> If these revisions address your concerns, we kindly request a reconsideration of the scores. Should you have any further questions, we are happy to assist.

---

> > ### Comment · Reviewer_GXeJ · 2025-04-02
> >
> > Thanks for the detailed reply. My concerns have been resolved. I increased the rate for this paper.

---

> > > ### Author Response · Authors · 2025-04-02
> > >
> > > Dear Reviewer GXeJ,
> > >
> > > We are very encouraged by your increased rate.
> > >
> > > We really enjoy the discussion, and thank you for your suggestion about open-ended generation with RSD.
> > >
> > > We believe this suggestion makes our work more solid and strong.
> > >
> > > Best!

---

### Official Review · Reviewer_p3vx · 2025-03-16

**Overall Recommendation:** 3

**Summary:**

This paper introduces a method to guide speculative decoding using a reward model. Unlike standard speculative decoding, where a larger model verifies the outputs of a smaller model, the proposed approach determines acceptance or rejection based on reward signals. Specifically, the authors design a series of weighting functions to normalize the reward as a probability that stochastically decides whether to accept or reject the generated responses. Experimental results demonstrate that this method maintains efficiency while improving reasoning performance.

**Claims And Evidence:**

Most of the claims in this paper are clear and well-supported. However, the statement that “High-quality tokens (e.g., those favored by a process reward) may still be rejected if their probabilities under the large model are too low” raises some concerns. This claim suggests a potential misalignment between the large model and the reward model. However, the proposed algorithm still trusts the large model’s output to correct responses rejected by the reward model (as shown in Eq. 1). This implies that while the large model has high precision, it may also exhibit a high false positive rate. Further justification is needed on this to explain why the reward model serves as a more reliable verifier than the large model’s likelihood or a combination of both.

**Essential References Not Discussed:**

No critical references known to me are missing.

**Experimental Designs Or Analyses:**

The experimental designs are overall sound. More details on how FLOPs are recommended to be included. It is also suggested to include more efficiency metrics, such as throughput (tokens per secs).

**Methods And Evaluation Criteria:**

The proposed methods are evaluated on 6 math reasoning benchmarks, which align with the major claim and the choice of reward models. The experiments also include a comparison on FLOPs.

**Other Comments Or Suggestions:**

Ln 30-31, “significant better” -> “significantly better”.
Ln 325-326, “majotity” -> “majority”

**Other Strengths And Weaknesses:**

Strength:
+ The overall writing quality is good, with clear mathematical formulations that effectively convey intuition to the reader. I found the results and discussions in Appendix C.2 particularly insightful, as they illustrate how RSD dynamically allocates computational resources based on task complexity.
+ The proposed approach is simple yet effective, offering a novel way to leverage reward models not only for enhancing reasoning accuracy but also for improving efficiency.
+ The experimental results are compelling, demonstrating that reward-guided decoding can even surpass the performance of the strongest large model.


Weakness:
- The theoretical results do not fully explain why RSD outperforms the large model.
- The motivation for why the reward model serves as a better acceptance criterion than the larger model is not entirely convincing. Since the algorithm ultimately relies on the larger model for corrections, it raises the question of why the reward model improves the verification process over large models’ likelihood. See “Claims And Evidence” for more details.
- Additionally, while speculative decoding in RSD is guided by process rewards, the baselines rely on outcome rewards. To ensure a fair comparison, it would be beneficial to include additional search-based baselines that also incorporate process rewards.

**Questions For Authors:**

The ablation results indicate that the threshold $\delta$ plays a crucial role in reasoning accuracy but is highly sensitive across different tasks. Given its definition in Proposition 2.3, can $\delta$ be estimated beforehand based on theoretical results, rather than relying solely on empirical tuning?

**Relation To Broader Scientific Literature:**

The proposed method has relevance to recent topics scaling inference-time compute to facilitate LLM reasoning, in which an LLM samples the solution according to a reward to do reasoning.

**Theoretical Claims:**

The theoretical results are partially checked (Props 2.1,2.2,2.3). It remains unclear what the “largest possible threshold that makes the function satisfy the constraint” refers to until I reach the appendix.

---

> ### Author Rebuttal · Authors · 2025-04-01
>
> **Q1.** PRM vs large models’ likelihood
>
> > The core issue is the misalignment between the large model (LM) and PRM, leading SD to reject high-quality tokens due to low LM likelihood. While the reviewer suggests this implies high FP rates from LM, these are often style-/format-related, not correctness errors. LMs generalize better but are NOT trained for correctness. In contrast, PRMs are explicitly trained to identify correctness, making them more suitable for detecting errors, while LM likelihood would point out both style inconsistency and errors. Thus, our algorithm uses LM guidance only after PRM rejection, leveraging both PRM's specialized correctness identification and LM’s strength in generalizing better completions. Using both for rejection is also promising; we leave it for future work as noted in Sec 2.4.
>
> $~$
>
> **Q2.**  Calculation of FLOPs.
>
> > As described in Sec 3.3, we follow the standard FLOPs approximation for transformer models with N parameters, i.e., approximately 2N FLOPs per inference token, as adopted in prior works [1,2]. We provide a detailed example below:
> > The FLOPs calculation for RSD (7B/72B/7B) is presented in the table below.
> >
> ||**Target**|**Draft**|**PRM**|
> |--------------|--------------|--------------|--------------|
> |Model Size| 72B | 7B|7B|
> |Tokens/Question| 67 | 396|18|
> |TFLOPs|9.648|5.544|0.252|
> >
> > [1] Scaling laws for neural language models.
> >
> > [2] Beyond chinchilla-optimal: Accounting for inference in language model scaling laws.
>
> $~$
>
> **Q3.**  More efficiency metrics (throughput).
>
> > We measure the througput and provide the following table. We use batch size 256 and MATH500.
> > The observation is similar to Figure 4: RSD is faster than SD.
>
>  | Method | Throughput |
>  | --- | --- |
>  | Single Draft (1.5B) | 1.00 $\times$ (4697 tokens/s) |
>  | Single Target (7B) | 0.70 $\times$ |
>  | SD(1.5B/7B) | 0.83 $\times$ |
>  | RSD(1.5B/7B/1.5B) | 0.91 $\times$ |
>  | RSD(1.5B/7B/7B) | 0.85 $\times$ |
>
> $~$
>
> **Q4.** Search-based baselines
>
> > We compare RSD to the search-based baselines (Process Best-of-N and Beam Search) in Table 3, where RSD significantly outperforms the search-based baselines that only utilize a draft model and PRM. It shows the importance of involving a larger model in the reasoning path for correction.
>
> $~$
>
> **Q5.** Why RSD outperforms the large model?
>
> >**Proposition (Improved Expected Reward via RSD):**
> Assume that for each decoding step the reward function $r(y|z)\in[0,1]$ and consider $\omega(r) = \mathbf{1}(r \geq \delta)$ for some threshold $\delta \in [0,1]$. Define the RSD mixture distribution as $ \mathbf{P} _ {\text{RSD}}(y| z) = \omega(r(y| z)) \mathbf{P} _ m(y | z) + \nu\mathbf{P} _ M(y| z), $
> where $\nu = 1 - \mathbf{P} _ {y\sim \mathbf{P} _ m}\{r(y| z) \geq \delta\}$.
> The expected reward of the RSD distribution is
> $$\mathbf{E} _ {\mathbf{P} _ {\text{RSD}}}[r(y|z)] = \alpha\,\mathbb{E} _ {\mathbf{P}_m}[r(y| z)| r(y| z) \geq \delta] + (1-\alpha)\,\mathbb{E} _ {\mathbf{P}_M}[r(y| z)],$$
> with $\alpha = \mathbf{P} _ {y\sim \mathbf{P}_m}\{r(y| z) \geq \delta\}.$
>
> >The RSD distribution outperforms the target model $\mathbf{P} _ M$ (i.e., $\mathbf{E} _ {\mathbf{P} _ {\text{RSD}}}[r(y| z)] > \mathbf{E} _ {\mathbf{P} _ M}[r(y| z)]$) if and only if
> $$\mathbf{E} _ {\mathbf{P}_m}[ r(y| z) | r(y| z) \geq \delta] > \mathbf{E} _ {P_M}[r(y| z)].$$
>
> >Thus, RSD yields a higher expected reward than $\mathbf{P}_M$ provided that the subset of tokens generated by the draft model $\mathbf{P}_m$ (i.e. those with $r(y| z) \geq \delta$) has a higher reward than $\mathbf{P}_M$. If one can choose $\delta$ so that only high-quality tokens from the draft model are accepted, then the mixture of these with the fallback tokens from $\mathbf{P}_M$ lifts the overall expected reward of RSD above that of the large model alone.
>
> >This result offers a theoretical explanation for why RSD can outperform the large model: it leverages the fact that, when the draft model’s top-performing outputs (as measured by $r$) are even better on average than the target model’s outputs, selectively accepting these outputs increases the overall quality.
>
> >In summary, for RSD to be advantageous over $\mathbf{P}_M$, one must choose a threshold $\delta$ such that:
> $$\mathbf{E} _ {\mathbf{P}_m}[r(y| z)| r(y| z) \geq \delta] > \mathbf{E} _ {\mathbf{P}_M}[r(y| z)].$$
>
> >Under this condition, RSD leads to an improved expected reward compared to $\mathbf{P}_M$.
>
> $~$
>
> **Q6.** Estimation of δ
>
> >**A6.**  As noted in Prop. 2.3, $\delta$ corresponds to a quantile of the reward distribution under a given compute budget. While theoretical guidance informs its range, estimating the exact quantile requires access to empirical reward statistics -- hence, practical tuning (e.g., grid search) is necessary and suggested by our analysis. Moreover, Table 2 demonstrates that performance is not highly sensitive to $\delta$, suggesting our method is robust across a reasonable range and not reliant on fine-tuning.

---

### Decision · Program_Chairs · 2025-05-01

**Decision:**

Accept (poster)

**Comment:**

This paper introduces process reward models (PRMs) into the speculative decoding framework. Instead of requiring the large model to strictly verify the small model’s output, a reward model is used to judge whether the small model’s output is acceptable. This allows token-level differences, reducing the intervention frequency from the large model. As a result, inference becomes more efficient without sacrificing reasoning quality. The method achieves up to 4.4× speedup compared to using the large model alone, and improves accuracy by up to 3.5 points over parallel decoding with the small model on reasoning tasks.

All reviewers are in favor of acceptance. The paper presents well-designed experiments and introduces insightful ideas. I recommend acceptance.

One minor suggestion, raised by reviewers p3vx and XUtc, concerns the criterion for selecting the reward threshold $\delta$. The paper would benefit from a clearer explanation of how $\delta$ is chosen, along with an analysis of its impact across broader domains to support practical applications.